

# Evaluating Simplified Chemical Mechanisms within CESM Version 1.2 CAM-chem (CAM4): MOZART-4 vs. Reduced Hydrocarbon vs. Super-Fast Chemistry

Benjamin Brown-Steiner[1,2,3], Noelle E. Selin[3,4,5], Ronald Prinn[2,3,5], Simone Tilmes[6], Louisa Emmons[6], Jean-François Lamarque[6], Philip Cameron-Smith[7]

[1]Now at Atmospheric and Environmental Research, 131 Hartwell Avenue, Lexington, MA 02421-3126
[2]Center for Global Change Science, Massachusetts Institute of Technology, 77 Massachusetts Ave, Cambridge, MA 02139
[3]Joint Program on the Science and Policy of Global Change, Massachusetts Institute of Technology, 77 Massachusetts Ave,
Cambridge, MA 02139
[4]Institute for Data, Systems, and Society, Massachusetts Institute of Technology, 77 Massachusetts Ave, Cambridge, MA 02139
[5]Department of Earth, Atmospheric, and Planetary Sciences, Massachusetts Institute of Technology, 77 Massachusetts Ave, Cambridge, MA 02139
[6]Atmospheric Chemistry Observations and Modeling Lab, National Center for Atmospheric Research, 3450 Mitchell Lane, Boulder, CO 80301
[7]Lawrence Livermore National Laboratory, 7000 East Ave, Livermore, CA 94550

*Correspondence to*: Benjamin Brown-Steiner (bbrownst@aer.com)





**Abstract.** While state-of-the-art complex chemical mechanisms expand our understanding of atmospheric chemistry, their sheer size and computational requirements often limit simulations to short length, or ensembles to only a few members. Here we present and compare three 25-year offline simulations with chemical mechanisms of different levels of complexity using

CESM Version 1.2 CAM-chem (CAM4): the MOZART-4 mechanism, the Reduced Hydrocarbon mechanism, and the Super-Fast mechanism. We show that, for most regions and time periods, differences in simulated ozone chemistry between these three mechanisms is smaller than the model-observation differences themselves. The MOZART-4 mechanism and the Reduced Hydrocarbon are in close agreement in their representation of ozone throughout the troposphere during all time periods (annual, seasonal and diurnal). While the Super-Fast mechanism tends to have higher simulated ozone variability and

differs from the MOZART-4 mechanism over regions of high biogenic emissions, it is surprisingly capable of simulating ozone adequately given its simplicity. We explore the trade-offs between chemical mechanism complexity and computational cost by identifying regions where the simpler mechanisms are comparable to the MOZART-4 mechanism, and regions where they are not. The Super-Fast mechanism is three times as fast as the MOZART-4 mechanism, which allows for longer simulations, or ensembles with more members, that may not be feasible with the MOZART-4 mechanism

given limited computational resources.



**Copyright Statement**

- Authors retain the copyright of the article. Regarding copyright transfers please see below.

- Authors grant Copernicus Publications an irrevocable non-exclusive license to publish the article electronically and in print format and to identify itself as the original publisher. Authors grant Copernicus Publications commercial rights to produce hardcopy volumes of the journal for sale to libraries and individuals.

- Authors grant any third party the right to use the article freely as long as its original authors and citation details are identified.

- The article is distributed under the Creative Commons Attribution 4.0 License. Unless otherwise stated, associated published material is distributed under the same license.

- This manuscript has been co-authored by Lawrence Livermore National Security, LLC under Contract No. DE-AC52-07NA2 734-I with the US. Department of Energy. The United States Government retains, and the publisher, by accepting the article for publication, acknowledges that the United States Government retains a non-exclusive, paid-up, irrevocable, world-wide license to publish or reproduce the published form of this manuscript, or allow
15
others to do so, for United States Government purposes.



# 1 Introduction

    The anthropogenic influence on atmospheric chemistry is apparent at all spatial and temporal scales: human emissions have impacted local and very short-lived species (e.g. OH, see Prinn et al., 2001), very long-lived greenhouse gases (e.g. Collins et al., 2006) and everything in between (e.g. Baker et al., 2015; Solomon et al., 2016). Over the past decades, all

three branches of modern atmospheric chemistry research (Abbatt et al., 2014) – observations, laboratory analysis, and modeling – have increased in both their sophistication and their capability to explain the chemistry of our atmosphere. However, while observational networks have significant growth potential (e.g. Sofen et al., 2016), and laboratory analysis still has significant challenges to overcome (Bocquet et al., 2015; Burkholder et al., 2017), chemistry modeling efforts are finding their growth potential is limited by the level of chemical complexity that can be included in models due to the

constraint of the computational capabilities of even state-of-the-art supercomputers (Stockwell et al., 2012). Simulations that attempt to include all known species and reactions, such as the National Center for Atmospheric Research (NCAR) Master Mechanism (Madronich and Calvert, 1989, Aumont et al., 2000) or the Leeds Master Chemical Mechanism (Jenkin et al, 1997; Saunders et al., 2003), and even some species and reactions that have not been tested in any laboratory (e.g. Aumont et al. 2005; Szopa et al., 2005), are often limited to box-model level analysis (e.g. Emmerson and Evans, 2009; Squire et al.,

2015). Modeling efforts that simulate regional- and global-scale atmospheric chemistry are forced, out of practical necessity, to utilize simplified, reduced form, and parameterized chemistry in order to address the large spatial and long temporal scales needed for much policy-relevant research.

    Historically, as computational capacity has increased, modeling efforts have tended to maximize model resolution and complexity. This limits the capability to perform multi-scenario or multi-model ensembles to institutions with access to

significant computational capabilities and storage. One way to increase the number of scenarios, or members, in an ensemble is to reduce the complexity of the chemical mechanism. This selection of a reduced-form chemical mechanism for different applications, and the advantages of the increased computational efficiency of a simplified mechanism, is the main focus of this paper. While there is a long history of publications (see Dodge, 2000) that compare different photochemical mechanisms within box models (e.g. Milford et al., 1992; Jimenez et al., 2003; Emmerson and Evans, 2009; Knote et al., 2015), studies

that compare multiple mechanisms within a single 3D global model are rare (e.g. Squire et al., 2015). This study examines three chemical mechanisms within the Community Earth System Model Community Atmosphere Model with Chemistry Version 1.2 (CESM1.2 CAM-chem; Lamarque et al., 2012) framework: the MOZART-4 mechanism, the Reduced Hydrocarbon mechanism, and the Super-Fast chemical mechanism (described in Section 2), which is one of the simplest representations of atmospheric chemistry in the published literature.

This study examines the trade-offs and possibilities that arise from selection of a chemical mechanism that is simple enough to be computationally efficient – and thus capable of long simulations, or large ensembles at the global scale – as well as sophisticated enough to simulate the major features of tropospheric chemistry at the local and regional scale. Many climate studies include little to no chemistry, or prescribed chemistry, even though chemistry-climate feedbacks are well established to impact global and regional climate (e.g. Marsh et al. 2013; Fiore et al., 2015). Indeed, coarse grid (2˚x2.5˚)





chemistry-climate studies which conduct 1,000 or more years of simulations using complex chemistry are notable in their rarity (for notable exceptions, see Barnes et al., 2016 and Garcia-Menendez et al., 2015, 2017). This paper focuses on three primary lines of inquiry focusing on tropospheric ozone. First, what is lost or gained with the selection of a simplified chemical mechanism within a global model? Second, what is the nature of the uncertainties that arise with the selection of a

particular chemical mechanism? And third, what are the tradeoffs that researchers make, either intentionally or tacitly, when they apply that mechanism within a particular modeling framework? We focus this study on the short-lived gaseous species, in particular ozone and its precursors, that influence both the daily exposure of humans to pollutants as well as the decadal-scale global climate system. We focus primarily on a computationally efficient simulation of tropospheric gaseous chemistry within a single modeling framework, and leave further analysis of other aspects of atmospheric chemistry to future studies.

In Section 2, we describe the modeling framework, and describe each of the three aforementioned chemical mechanisms, including a detailed description and history of the Super-Fast mechanism, as it is not reported elsewhere in the literature, and the simulations and observations we use for comparison. In Section 3 we present spatial and temporal results, as well as compare various metrics of chemical accuracy. In Section 4, we explore the nature and the morphology of the chemical uncertainties, and the particular tradeoffs that are made by the selection of a single mechanism when faced with

limited computational resources. We draw conclusions in Section 5.

## 2 Methods

    Our analysis focuses on characterizing the ozone chemical uncertainties within a global chemistry model. We examine the morphology of the chemistry system, focusing specifically on the means, standard deviations, and variability (defined

here as the standard deviation divided by the mean). We also include characterizations of the correlation of the ozone time series with the observations and of the extreme values (in particular the $90^{th}$ and $99^{th}$ percentiles) of the ozone distribution.

### 2.1 CESM1.2 CAM4-chem Simulations

    The CESM1.2 CAM4-chem model (Tilmes et al., 2015; 2016) is a chemistry-climate model developed at the

National Center for Atmospheric Research (NCAR) with other collaborators, including the U.S. Department of Energy. It has been utilized extensively in the Atmospheric Chemistry and Climate Intercomparison Project (ACCMIP) (Lamarque et al., 2013 and references therein), the Chemistry Climate Model Initiate (CCMI) (Morgenstern et al., 2017) and for a wide range of atmospheric chemistry research. We conduct our simulations using CESM CAM4-chem version 1.2 with the MOZART-4 chemical mechanism based on Emmons et al. (2010) with updates described in Tilmes et al. (2015), the

Reduced Hydrocarbon mechanism (Houweling et al., 1998) as adapted to the CESM CAM-chem framework by Lamarque et al. (2008, 2010), which has a reduced form representation of hydrocarbon chemistry, and the Super-Fast mechanism (Cameron-Smith et al., 2006; Lamarque et al. 2013). Hereafter we will refer to these three mechanisms as MO, RH, and SF, respectively.





For meteorology we used the Modern-Era Retrospective analysis for Research and Applications (MERRA) reanalysis product (Rienecker et al., 2011) for 25 years (1990 – 2014), with a 50-hour Newtonian relaxation timing (roughly 1% nudging every 30 minutes). All simulations are at 1.9˚x2.5˚ resolution. Aerosols were represented by the bulk aerosol model (BAM) in the MO and RH mechanisms and is optional for the SF mechanism. We keep anthropogenic emissions

constant at year-2000 from the CCMI database (Lamarque et al., 2012) and include linearized chemistry for ozone in the stratosphere (McLinden et al., 2000; Hsu and Prather, 2009), and prescribe the concentration of other tracers above 50 hPa. We use an online biogenic emissions model (MEGAN; Guenther et al., 2012), and prescribed sea ice and sea surface temperatures. With the exception of a remapping of the MOZART species to the Reduced Hydrocarbon species (Supplemental Table S1), all parameterizations other than the chemical mechanism are identical between the three

simulations, and thus any differences are due to differences among the mechanisms themselves. Ozone dry deposition was done as described in Val Martin et al. (2015). Because we run with prescribed meteorology, we do not include internal chemical feedback to the weather and climate other than that incorporated into the MERRA meteorology itself. All of these mechanisms can also be run with meteorology calculated internally by the CESM model, but since such simulations utilize a different number of vertical levels than simulations with prescribed meteorology, comparing to simulated meteorology runs

is not straightforward, and so is omitted from the present study.

### 2.2 Mechanisms

Table 1 summarizes the characteristics of the three chosen mechanisms.

### 2.2.1 MOZART-4 (MO)

The Model for Ozone and Related Chemical Tracers, version 4 (MOZART-4) mechanism (Emmons et al., 2010) is the standard tropospheric chemical mechanism used within the CESM CAM-chem framework (Tilmes et al., 2015; 2016). It has been used in many model inter-comparison projects (e.g. Lamarque et al., 2013; Emmons et al., 2015), and extended to tagged tracer chemistry (Emmons et al., 2012).

### 2.2.2 Reduced Hydrocarbon (RH)

The Reduced Hydrocarbon (RH) chemical mechanism (Houweling et al., 1998; Lamarque et al., 2010) is a reduced-form mechanism based on the Carbon Bond Mechanism 4 (CBM-4) (Gery et al., 1989). The CBM-4 was developed to simulate polluted regional chemistry, and the RH mechanism updated and expanded this mechanism to also be capable of

simulating background low-$NO_x$ conditions (Houweling et al., 1998). As described in Houweling et al. (1998), the original RH mechanism has 30 tracers and 68 total reactions. It has been used extensively in model inter-comparisons (e.g. Pöshl et al., 2000) and is generally considered a satisfactory reduced hydrocarbon mechanism (e.g. Hauglustaine et al., 1998; Wang





and Prinn, 1999; Granier et al., 2000; Pfister et al., 2014). Lamarque et al. (2008) incorporated the RH mechanism into the CESM CAM-chem framework with a few updates, and Lamarque et al. (2010) expanded it to 89 (to include the bulk aerosol model species) tracers and 202 total reactions. As the lumping of alkanes and alkenes in RH differs from the MO mechanism, a mapping between the differently aggregated species is necessary (see Supplemental Table S1).

For this work, we modified the RH mechanism to remove many of the tracers and reactions that are pertinent primarily to stratospheric chemistry (as introduced in Lamarque et al., 2008) since these simulations include specified long-lived stratospheric species ($O_3$, $NO_x$, $HNO_3$, $N_2O$, $N_2O_5$) as in MOZART-4 (Emmons et al., 2010). However, the unmodified RH mechanism can be run with the more complex stratospheric chemistry, but at a significant additional cost. This is not considered in this paper to allow a better comparison between the tropospheric-only mechanisms. The modified RH
mechanism, which shows only minor differences in the simulated surface ozone concentration from the complete mechanism (not shown), contains 65 tracers and 127 reactions. This RH mechanism runs approximately twice as fast than the MO mechanism under our current configuration (Table 1).

### 2.2.3 Super-Fast (SF)

The Super-Fast (SF) mechanism is a highly simplified chemical mechanism designed to efficiently simulate background tropospheric ozone chemistry (Cameron-Smith et al., 2006, and supplementary material of Lamarque et al., 2013). It includes 15 chemical tracers with 6 photolysis reactions and 24 gas phase reactions, making it the simplest chemical mechanism to be included as a member of the ACCMIP ensembles (Lamarque et al., 2013). It was developed by the Lawrence Livermore National Laboratory (LLNL) and the detailed chemical mechanism within the CESM code has not
previously been published, so we include a description here and in our Supplementary Material.

Supplemental Table S2 summarizes the SF mechanism photolysis and gas-phase reactions, which consist of a basic methane oxidation scheme ($CH_4$, $CH_3O_2$, $CH_3OOH$, $CH_2O$, and CO), with basic oxidant chemistry (OH and $O_3$), along with simple sulfur chemistry (dimethyl sulfide (DMS), $SO_2$, and $SO_4$) and a single biogenic hydrocarbon species, isoprene (ISOP), with two oxidant pathways: ISOP + OH and ISOP + $O_3$. For reactions iii, vi, 10, 11, and 15 (Table S2), it is assumed
that their products O, H, and $CH_3OH$ are instantaneously converted to their ultimate products, $O_3$, $HO_2$, and $HO_2$, respectively. Nitric acid chemistry is limited to two reactions, one of which requires a heterogeneous reaction parameterization. Sulfur chemistry is limited to four reactions. Isoprene chemistry is highly parameterized. The reaction of isoprene with OH is based on the net effect of the reaction in the University of California Irvine (UCI) model (Wild and Prather, 2000), namely: ISOP + 2.5*OH → 2*$CH_2O$. This particular parameterized reaction, which during the original
implementation required a negative coefficient for the OH product, is not standard within the CESM chemical modeling framework and cannot be handled by the solver, so the equivalent triple reaction formulation of 21a, 21b, and 21c is required. The oxidation of isoprene by ozone is a simple parameterization (resulting in fractional production of only the species that already exist in the mechanism as part of the methane oxidation scheme: $CH_2O$, $CH_3O_2$, $HO_2$, and CO) derived



from the net effect of the isoprene/ozone oxidation pathways from the full LLNL-IMPACT model (Rotman et al., 2004) and was included specifically to improve the simulation of surface ozone chemistry (Cameron-Smith et al., 2009).

Much of the simplicity within the SF mechanism comes from what it does not include. Carbon chemistry is limited to the five single-carbon species used in the simple methane oxidation scheme, plus isoprene. There is no PAN (peroxy
acetyl nitrate) or ammonia, and hence no nitrogen aerosols, although $HNO_3$ is created in reaction 8 and 16. These all impact ozone chemistry, but the inclusion of additional hydrocarbon, aerosol, or heterogeneous chemistry would introduce significant additional computational costs (similar to the more complete mechanisms). There are no halogen species, since this would require the inclusion of a significant number of additional chemical tracers, and as such there is no capability to describe the polar ozone hole phenomenon within the mechanism (Cameron-Smith et al., 2006), so it is implemented within
Linoz using the simple loss parameterization of Cariolle, et al., 1990. The greatest simplifications in the SF mechanism arise from compacting all of the non-methane hydrocarbon chemistry (NMHC) into two isoprene reactions, and thus there is none of the complex chemistry that is required to adequately represent ozone chemistry in highly polluted regions. The simplicity of the SF mechanism allowed us to perform two short simulations in which we added reduced-form PAN and $N_2O_5$ chemistry (individually, and in conjunction) from the MOZART-4 mechanism into the SF mechanism, which we use as a
demonstration of the type of sensitivity tests that are possible with the SF mechanism. This type of quick sensitivity test would be significantly more difficult with the more complex mechanisms, given the complexity of PAN and $N_2O_5$ chemistry.

The SF mechanism has been included in several model inter-comparison projects. We include an expanded review in the Supplemental Material. Unfortunately, the SF mechanism only simulates sulfate aerosol, and so was unable to be compared to the aerosol simulations of the other ACCMIP members (Lamarque et al., 2013). The SF simulations within
ACCMIP demonstrated lower rates of ozone chemistry and deposition resulting in a low ozone burden bias and a high ozone lifetime bias (Young et al., 2013), and while projected changes in ozone radiative forcing fell within the ACCMIP range, the historical changes did not (Stevenson et al., 2013). Human health analysis with the SF simulations fell within the range of the other ACCMIP members (Silva et al., 2013; 2016; 2017). Squire et al. (2015) compared SF to more complicated isoprene schemes, and concluded that including the SF mechanisms is preferable to neglecting chemistry entirely, although there are
biases in regions of high biogenic chemistry. Finally, Schnell et al. (2015) compare seasonal and diurnal cycles to other mechanisms, and the SF mechanism simulates high ozone events in the springtime, and that the SF mechanism outperforms others when compared to the observed summertime diurnal cycle.

### 2.3 Computational Requirements

The computational requirements of MO, RH, and SF as simulated on the NCAR Cheyenne supercomputer are summarized in Table 1. The computational cost results from both the chemical solver and the advection of the chemical tracers within CAM-chem. No load balancing was conducted, which could potentially increase the efficiency of the RH and SF mechanisms. The CESM1.2 CAM-chem model run with the SF mechanism is roughly three times faster than a run with





the MO mechanism when BAM aerosols are included (which we do not examine in this present study), and a gas-phase-only simulation with the SF mechanisms increases the speeds to nearly 4 times as fast. The RH mechanism is roughly twice as fast as the MO mechanism. At higher spatial resolutions, and the computational advantage of the SF mechanism over the more complex MO and RH schemes is likely to increase, since advection of tracers typically becomes a larger fraction of the total model run-time.

### 2.4 Observations

The ozone observational databases are of two types: the global database is ozonesonde data compiled from Tilmes et al. (2012) while the US database comes from the EPA Clean Air Status and Trends Network (CASTNET), which has more than 90 surface observational sites within the United States and has been collecting surface meteorological and chemical data since 1990 (CASTNET, 2016 and https://www.epa.gov/castnet). We used data from all sites that reported complete ozone data from each year, after removing data that the CASTNET database marked as invalid. The number of sites that matched these criteria varied from year to year, but generally we have between 55 and 94 sites throughout the 1990 – 2014 period. The CASTNET observational network is located primarily in rural sites, and thus is a reasonable comparison to coarse grid cell model output. In order to compare to the CESM CAM-chem simulations, which has no emissions trend, we have detrended the CASTNET data for each region using a simple linear regression. Regional averaging is first done by averaging all observational sites within a single 1.9˚x2.5˚ grid cell, and then averaged to the larger regions as needed. We also compare to ozone precursor species observations from Tilmes et al., (2015).

### 3 Results

### 3.1 Spatial Comparisons

The spatial distribution of ozone and related species between the three mechanisms are compared in Figure 1. Taylor-like diagrams comparing results to ozonesondes over different global regions are provided in Figure 2, and comparisons to aircraft observations in Figure 3. Globally averaged surface Daily Maximum 8-Hour (DM8H) $O_3$ is consistent across all mechanisms (Table 2) with the largest spatial differences (especially with the SF mechanism) noted over regions of intense biomass burning or biogenic emissions, such as equatorial Africa and South America, as well as over northern hemisphere oceans within SF (Figure 1). Surface CO mixing ratios show small regional differences between MO and RF, while $NO_x$ mixing ratios show very small and highly localized differences (Figure 1). All three mechanisms tend to have low CO biases over much of the northern hemisphere, with SF showing the largest bias. This coincides with starkly higher $NO_x$ mixing ratios in the northern hemisphere (Figure 1, Figure 3), especially in the winter and spring seasons. This is explored in more detail below.

Zonal profiles (Figure 4) show that ozone is similar among all mechanisms for all seasons, especially in the lower troposphere. Compared to the MO mechanism, the SF mechanism simulates higher northern hemisphere ozone in the winter,



and lower in the summer. Both the RH and SF mechanisms simulate lower CO mixing ratios than the MO mechanism in both the summer and winter, with the SF mechanism diverging the most in the northern hemisphere in the summer. The SF mechanism also simulates higher $NO_x$ in the northern hemisphere winter, which (as we explore below) may in part be due to the lack of PAN chemistry.

5        At the largest spatial scales, all three mechanisms predict similar levels of surface ozone (Figure 5, Table 2), with global surface ozone estimates of $32.6 \pm 0.93$, $33.9 \pm 0.98$, and $31.5 \pm 1.12$ ppb for MO, RH, and SF, respectively. Even at the Continental US scale, all three mechanisms estimate similar surface DM8H $O_3$ values ($56.7 \pm 3.08$, $57.7 \pm 3.23$, and $53.4 \pm 3.59$ ppb for MO, RH, and SF, respectively), which are consistent with the CASTNET observations of $56.1 \pm 5.65$ ppb. However, within the Northeastern US, the well-known high bias is apparent ($74.4 \pm 11.4$, $76.0 \pm 11.9$, $72.6 \pm 14.5$ ppb for

MO, RH, and SF, respectively, while the CASTNET observations are $57.4 \pm 7.42$ ppb). The MO and RH mechanisms are nearly identical at all spatial scales, while the SF mechanism simulates larger DM8H $O_3$ variability, especially at individual grid cells within the Eastern US. Taking into account the model ozone biases, the SF is a better characterization of the ozone distribution (as compared to CASTNET) for almost every spatial scale examined within the US. Indeed, in the Southeastern US, where we expect SF to perform poorly due to the simplified biogenic species chemistry, we actually find that the SF

estimates the shape of the high ozone tail better than either MO or RF: CASTNET estimates at an individual grid-cell, that the 99[th] percentile for DM8H $O_3$ is 18% higher than the 90[th] percentile (Table 2), and while MO and RH estimate only 14% higher and 14% higher, respectively, the SF estimates 29% higher. In Section 4, we explore some of the implications of these differences, and in particular whether the biases within the SF mechanism are of the same magnitude as some of the biases within the MO and RF.

20        Figure 6 explores this finding, which plots the percentage difference between the 99[th] and the 90[th] percentile ozone as the length of the time series included grows. We note that: (1) it takes between 5 and 10 years before a consistent and stable estimate emerges with each simulation, indicating that simulations less than 10 years may be inadequate for comparisons between chemical mechanisms; (2) the CASTNET observations have a transient estimate, most notably in the Southeastern US, which indicates a divergence of the 99[th] and the 90[th] percentiles (i.e. a lengthening of the upper tail) that is

not seen in the simulations; and (3) the SF mechanism is inconsistent with the MO and RH mechanisms, which are nearly identical, but the SF mechanism estimate is also closer to the CASTNET estimate in the Midwestern and Southeastern US. Whether this is the result of fortunate biases within the SF mechanism or an implication that the more complex chemistry within the MO and RH mechanisms are underestimating the length of the ozone tail requires further study.

       However, while the SF mechanism performs as well as, or better than, the MO and RH mechanisms in certain

regions, there are many regions – especially in the northernmost latitudes over land, and over equatorial land masses – where the SF mechanism is far less capable at simulating surface ozone than either the MO or RH mechanisms. Figure 7 plots $R^2$ values for the DM8H $O_3$ JJA time series (1990 – 2015) at every grid cell between the MO mechanism and both RH and SF, and it is clear that the RH mechanism has very high $R^2$ values ($R^2 > 0.75$) over much of the globe. And while the SF





mechanism has large $R^2$ values over many regions – in particular the extratropics – over the equatorial regions, and especially over land, $R^2$ values drop below 0.5 and even 0.25.

### 3.2 Seasonal and Diurnal Comparisons

The seasonality of surface ozone is similar among all three mechanisms at the regional-scales (Figure 8), although differences occur at both the largest and smallest scales: (1) the SF mechanism simulates a dual-peaked maximum in surface ozone averaged at the global scale, a phenomenon also noted by Schnell et al. (2015); (2) this dual-peaked maximum is still apparent at the regional scales, although to a much lesser degree; and (3) the RH mechanism has a dual-peaked maximum over portions of the Southeastern US. The seasonal patterns for CO and $NO_x$ are consistent across all models, although CO is

lower in both RH and SF than in MO for all seasons. RH and MO $NO_x$ levels are nearly identical, but SF simulates higher values for $NO_x$ in all seasons, and particularly in the winter and spring seasons, as already noted. $HO_x$ and isoprene seasonality is consistent across all mechanisms at most scales.

Diurnal cycles are compared for a single grid cell within the Central US in Figure 9. With the exception of isoprene within the SF mechanism, which does not adequately represent nighttime isoprene chemistry, the diurnal cycles are

comparable across all mechanisms for most species. The MO and RH mechanisms are nearly identical, with the exception of CO values, as already mentioned. The SF mechanism tends to show more extreme peaks in OH and $NO_x$, and lower levels of $O_3$, CO, $H_2O_2$, and $SO_4^=$ (Figure 9). Surface levels of $O_3$ and CO within the SF mechanisms are sensitive to the addition of PAN and $N_2O_5$ chemistry (the dotted lines in Figure 9), described below, although the sensitivity tends to be in the simulated magnitude and not the shape of the diurnal cycle.

Figures 8 and 9 also include two-year simulations (1990 – 1991, with year 2000 emissions) in which we included into the SF mechanism PAN and $N_2O_5$ chemistry taken (and reduced) from the MOZART-4 mechanism. We examine these mainly to demonstrate the potential for the modification of the SF mechanism to meet particular research needs. Largely, the addition of PAN chemistry (purple lines) results in more substantial changes to various species than the addition of $N_2O_5$ chemistry (orange lines), but their combined addition (green lines) slightly modifies the simulated large-scale values of $O_3$,

CO, $HO_x$, and isoprene. The addition of PAN chemistry brings the SF mechanism simulations closer to the MO mechanism for the $NO_x$ and $HO_x$ seasonal cycles (Figure 8), and the CO diurnal cycle (Figure 9), but at the expense of the global-scale capability to simulate ozone and isoprene. Additional tuning of the parameterized reactions 21 and 22 (Table S2) may be able to correct these errors. Sulfate aerosol in the SF mechanisms is notably lower than both the MO and RH mechanisms, which may result from the simple aerosol scheme within the SF mechanism.

### 3.3 Comparison to Observations

Figure 10 compares the model estimates of surface ozone to observations (ozonesondes and CASTNET observations) for different spatial regions, as well as to each other. Generally, all three mechanisms simulate less variability



over continental to global scale regions than the ozonesonde observations (Figure 10c,d,e) and show a high bias over many sites within North America, Europe, and Asia. Within the US, all mechanisms show a high bias in the Eastern US, and especially in the Northeastern US, but the variability is well-captured when compared to CASTNET (with slopes ranging from 0.61 – 1.24 in Figures 10f, g, and h). When compared to each other (Figures 10a,b,i,j), the RH mechanism and MO

mechanism are nearly identical. The SF mechanism, while comparable to the MO mechanism at many sites, shows greater divergence, overestimating values in many grid cells throughout the globe (Figure 10b) and both over- and underestimating within the US (Figure 10j). Taylor-like diagrams are plotted in Supplemental Figure S1 and show the close clustering of the MO and RH mechanisms, and that the SF mechanism differs from the observations at a similar magnitude than the MO and RH mechanism for some regions, but performs poorly in other regions (especially in the tropics, where tropospheric ozone is

underestimated with the SF mechanism).

**4 Discussion**

Our primary objective has been to determine what is lost (or gained) with the selection of a simplified chemical mechanism, which we summarize here. We mostly discuss the SF mechanism, as the tradeoffs with the RH mechanisms are

straightforward: we lose very little (Figure 10a and 10i) and gain about a 100% increase in simulation speed (Table 1). Many of the things that are lost with the use of the SF mechanism are expected: we lose the capability to directly simulate small-scale features of ozone chemistry in regions that depend strongly on complex biogenic chemistry. In particular, the equatorial landmasses – especially equatorial Africa and South America – are not well simulated (Figure 7). We also lose the capability to simulate some of the short-term features that require additional chemistry, such as the night-time behavior of

isoprene (Figure 9), or the cold season CO and $NO_x$ behavior (Figure 1 and 4). The addition of PAN and $N_2O_5$ chemistry do not rectify the nighttime behavior of isoprene (Figure 9), but do bring the cold-season simulated CO and $NO_x$ mixing ratios closer to the MO mechanism (Figure 8). These deficiencies may result from the highly parameterized biogenic chemistry within the SF mechanisms (Supplemental Table S2), although it may also result from the treatment of isoprene emissions, and future simulations will need to consider the trade-off between additional complexity and computational efficiency.

More surprisingly, there are several desirable capabilities that are not lost with the selection of the SF mechanism. For most regions, the selection of the SF mechanism does not degrade the estimate of surface ozone (both the magnitude and the variability), nor do we lose features of the daily variability that results from the meteorology. In many regions, and at many scales, we find that the selection of the SF mechanism introduces uncertainties that are smaller than the difference between the simulated and observed surface ozone mixing ratios (Figure 5). Surface layer ozone is adequately represented

over many regions in all seasons within the SF mechanism (Figure 8), despite the high CO and low $NO_x$ levels in the winter and spring seasons (Figure 4). For these seasons, the adequate ozone representation may be the result of compensating errors, and Schnell et al. (2015) previously found comparable cases where the SF mechanism outperforms more complex models, perhaps due to various sets of compensating biases or errors.





We now turn to the main question of this research: what do we gain when we select a simplified chemical mechanism? The primary thing we gain is the capability to simulate longer periods of time, or to include more members in an ensemble, in proportion to the simplicity of the mechanism. Our results show that, without any optimization of the code, the RH mechanism is ~ 100% faster than the MO mechanism, and the SF mechanism is up to 200% faster than the MO

mechanism (Table 1). We feel that the capability to run three SF simulations for the price of one MO simulation under different sets of initial conditions, for example, can extend the quantification of parametric uncertainties which is largely unavailable to the most complex and most computationally demanding mechanisms.

For instance, there are many research frameworks where the "three-for-one" advantage of the SF mechanism could be utilized with the MO mechanism to allow for an expanded exploration of parametric uncertainties that would not

otherwise be available with the MO mechanism alone. One simulation of a 5- or 10-year time slice with the MO mechanisms could be combined with three simulations of the SF mechanism, one matching the parameters of the MO mechanism (in order to provide a consistent baseline), and the other two exploring other parameter spaces (e.g. different initial conditions, or different emission scenarios). The establishment of a baseline comparison is particularly important, since the SF mechanism is a simplified mechanism, and should not be blindly trusted to reproduce the behavior of more complex

mechanisms. For example, if a research group is interested in precise estimates of ozone concentrations in regions where the biogenic influence is significant, the SF mechanism would prove insufficient. The RH mechanism may be sufficient, but the more modest increase in computational speed – a "two-for-one" advantage over the MO mechanism – may not be enough to justify the simulation. If, however, the phenomenon of interest can be shown to be within the SF mechanism capabilities (e.g. simulating regional-scale ozone, as shown in this paper), the "three-for-one" advantage of the SF mechanism is readily

apparent. The SF mechanism may be particularly desirable with chemistry-climate simulations at higher spatial resolutions.

In addition, the selection of a simplified mechanism allows for the capability to easily and efficiently test new forms and new representations of chemistry without the need to painstakingly update and test all possible interactions of any addition within a complex mechanism. For example, in this study, we added a simplified PAN and $N_2O_5$ representation to the SF mechanism (Figure 8 and 9) to see how it improves the simulations. This exercise offered a significant capability to

test, simulate, and further learn about improving atmospheric chemistry computations. This demonstrates that a hybrid approach (or tiered approach, as recommended in Uusitalo et al., 2015) – in which complex and trusted chemical mechanisms are used to evaluate simplified mechanisms that can run for longer periods or with increased ensemble members – has the potential to maximize computational capabilities and to get the most out of atmospheric chemistry modeling.

Furthermore, the selection of a simple chemical mechanism – especially when used in conjunction with more

complex mechanisms within a consistent modeling framework – allows for better quantification of the uncertainties, and the relative importance, of particular pieces of the chemistry. Here, for instance, the SF mechanism's representation of biogenic species chemistry is insufficient to adequately represent equatorial landmasses, but the reduced form RH mechanism is nearly as capable as the MO mechanism over most regions and most species. This begs the question: is there a representation of biogenic chemistry somewhere between the RH and the SF mechanisms that can approach the efficiency of the SF





mechanism and the accuracy of the RH mechanism? We hope that future research will address this question, as well as others, such as more globally oriented research pertaining to ozone budgets and the interaction between OH and $CH_4$ lifetime. In addition, comparisons of chemical mechanisms of different complexities, and particularly where the simplified mechanisms fail, could potentially identify regional chemical regimes. For instance, the SF mechanism cannot adequately

represent the chemistry of equatorial forests (Figure 7), and the spatial regions that fail to simulate ozone chemistry are similar to the spatial distribution of the tropical forest chemical regime identified in Figure 4 of Sofen et al. (2016), which utilized a statistical clustering technique to identify chemical regimes. Finally, the capability to examine atmospheric chemistry complexity in a step-wise fashion could also be utilized to bridge the gap between the most complex 3D chemical models and the more efficient models utilized by the Earth Models of Intermediate Complexity (EMIC) or Integrated

Assessment Model (IAM) communities.

## 5 Conclusion

In this study, we have compared three chemical mechanisms of different levels of complexity within the CESM CAM-chem framework. We conducted 25-year cycled emission simulations nudged to MERRA meteorology with the

standard tropospheric MOZART-4 (MO) mechanism of Emmons et al. (2010), the Reduced Hydrocarbon (RH) mechanism of Houweling et al. (1998), and the Super-Fast (SF) mechanism of Cameron-Smith et al. (2006). The RH mechanisms is roughly twice as efficient as the MO mechanism, and the SF mechanism is roughly three times as efficient as the MO mechanism, without any code optimization. As much as possible, we kept the parameterizations consistent across all mechanisms, although we had to remap some of the MO mechanism species to match up with the RH mechanism species.

We find that all three mechanisms successfully capture surface ozone values at the larger spatial scales, but at smaller spatial scales, and especially within the Northeastern US, all three mechanisms have surface ozone biases when compared to CASTNET observations, but that the mean values for all three mechanisms are consistent with each other at a variety of spatial scales. The SF mechanism simulations show larger ozone variability than the MO and RH simulations, although when normalizing the distributions to account for the known ozone biases, the SF mechanism represents the shape

and spread of the ozone distributions better than the MO or RH mechanisms, when compared to the CASTNET observations (Figure 5).

The RH mechanism is in close agreement with the MO mechanism for nearly every metric we examined, and any differences tend to be minor (both in magnitude and in spatial extent). The SF mechanism simulates higher $NO_x$ and lower CO than the MO mechanism, and the $NO_x$ deviations are particularly large in the winter season. In addition, the SF

mechanism deviates from the MO mechanism over regions of high biogenic emissions, such as equatorial Africa and South America. These large deviations within the SF mechanism are likely a result of the simplicity of the mechanism, and especially the lack of biogenic species chemistry beyond a single-species, two-reaction representation, as well as a lack of PAN and $N_2O_5$ chemistry (Figures 8 and 9). We also find that although the SF mechanism differs in the magnitude of the



estimated ozone from the other two mechanisms, the simulated ozone variability is similar in all three mechanisms (Figures 4 and 10).

We find that there are significant gains that can be realized by a research approach that utilizes simulations with both a complex and a simplified chemical mechanism where the complex mechanisms are used to provide a more-trusted

chemical result (especially for the mean values) and the simple mechanism could be used to efficiency simulate longer time periods to better understand the roles of meteorological variability. The capability of the SF mechanism to simulate adequate chemistry with interactive meteorology is not examined here, nor the coupling of the SF mechanism with modal aerosols, which is left for future research. These results encourage revitalizing or creating simplified chemical mechanisms within individual modeling frameworks, and examining the structural uncertainties that exist between different models with regards

to simplified chemical mechanisms.

Finally, we note that there are many inherent uncertainties associated with the use and comparison of chemical mechanisms and climate-chemistry simulations, many of which are inherited with the adoption of a particular model. The CESM CAM-chem model has been used extensively to examine a variety of climate and chemistry phenomena, and uncertainties that arise from the individual choices made during the historical development of this chemical model (see

Brasseur et al., 1998; Hauglustaine et al., 1998; Horowitz et al., 2003; Kinnison et al, 2007; Emmons et al., 2010) are still present in the CESM CAM-chem modeling framework, such as which scheme or parameterization was to be included and the specific metric and methodology of tuning the climate model to historical data (see Hourdin et al., 2017 and references therein). Future simulations using different model versions, or different choices of parameterizations, schemes, emissions, and other input datasets will need to examine the impact of those choices on the simulated chemical uncertainty and compare

these to the uncertainty that arises from the selection of the different chemical mechanisms presented here.



**Code Availability**

CESM CAM-Chem code is available through the National Center for Atmospheric Research /University Corporation for Atmospheric Research (NCAR/UCAR) website (http://www.cesm.ucar.edu/models/cesm1.2/), and this project made no code

5    modifications from the released model version.



**Data Availability**

The raw model output is archived on the NCAR servers, and processed data will be made available upon publication on a public recognized repository with a unique digital object identifier (doi).





**Supplemental Link**





**Author Contribution**

BBS prepared and ran the simulations and prepared the manuscript under direction and advisement of NS and RP. LE and ST aided in the development, preparation, and analysis of the simulations as well as reviewed the manuscript. JFL advised and aided in the Reduced Hydrocarbon simulation. PCS advised and aided in the Super-Fast simulation, and both reviewed

5   the manuscript.



**Competing Interests**

The authors declare that they have no conflict of interest.





**Acknowledgements**

This model development work was supported by the U.S. Department of Energy (DOE) Grant DE-FG02-94ER61937 to the MIT Joint Program on the Science and Policy of Global Change. The work of PC was supported through the Scientific Discovery through Advanced Computing (SciDAC) program funded by the DOE Office of Science, Advanced Scientific

5    Computing Research and Biological and Environmental Research, and was performed under the auspices of the DOE by Lawrence Livermore National Laboratory under Contract DE-AC52-07NA27344. Computational resources for this project were provided by DOE and a consortium of other government, industry, and foundation sponsors of the Joint Program. For a complete list of sponsors, see: http://globalchange.mit.edu. Additional computing resources were provided by the Climate Simulation Laboratory at NCAR's Computational and Information Systems Laboratory (CISL), sponsored by the National

10   Science Foundation and other agencies.  The National Center for Atmospheric Research is funded by the National Science Foundation. The authors would also like to thank Daniel Rothenberg for efficient processing of the ozone files.

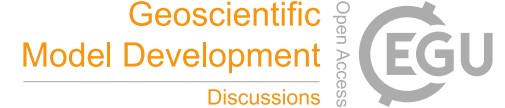

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





| Abbreviation | MO | RH[*] | SF[**] |
|---|---|---|---|
| Full Name | MOZART-4 | Reduced Hydrocarbom | Super-Fast |
| Primary Citation | Emmons et al. (2010) | Houwelling et al. (1998) | Cameron-Smith et al. (2006) |
| Total Tracers | 103 | 65 (89) | 15 |
| Total Reactions | 212 | 127 (202) | 30 |
| Reactions in NHMC Chemistry | 108 | 28 | 2 |
| core hours / simulated year | 615 | 319 | 165 (204) |
| simulated years / day | 2.5 | 4.8 | 9.3 (7.5) |
| Efficiency (compared to MO) | 1.0 | 1.9 | 3.7 (3.0) |

\*: unmodified RH listed in the parenthesis
\*\*: SF + Bulk Aerosol Model (BAM) included in parenthesis

**Table 1:** Summary and comparison of the MOZART-4 (MO), Reduced Hydrocarbon (RH), and Super-Fast (SF) mechanisms included in this paper. All runs were conducted on the NCAR Cheyenne system with 64 CPUs on 2 nodes without any load optimization, and the values in this table represent the cost of the entire CESM CAM-chem model, not just the chemistry component. In this study, we removed many stratospheric species (see text), so we include both the modified and unmodified (in parenthesis) RH mechanisms. The MO and RH mechanism include BAM.



| | | [ppbv] Mean | [ppbv] Median | [ppbv] Standard Deviation | [%] Variability | [ppbv] 90th Percentile | [ppbv] 99th Percentile | [ppbv] 99th – 90th Percentile | [%] |
|---|---|---|---|---|---|---|---|---|---|
| Global | MO | 32.6 | 32.6 | 0.93 | 2.86 | 33.8 | 35.3 | 1.52 | 105 |
| | RH | 33.9 | 33.9 | 0.98 | 2.90 | 35.2 | 36.4 | 1.25 | 104 |
| | SF | 31.5 | 31.4 | 1.12 | 3.57 | 33.0 | 34.1 | 1.07 | 103 |
| CO US | CASTNET | 56.1 | 55.8 | 5.65 | 10.1 | 63.4 | 71.0 | 7.60 | 112 |
| | MO | 56.7 | 56.6 | 3.08 | 5.43 | 60.6 | 64.4 | 3.82 | 106 |
| | RH | 57.7 | 57.6 | 3.23 | 5.60 | 61.8 | 65.9 | 4.10 | 107 |
| | SF | 53.4 | 53.3 | 3.59 | 6.72 | 57.9 | 62.8 | 4.87 | 108 |
| E US | CASTNET | 56.4 | 56.0 | 6.41 | 11.4 | 64.5 | 73.3 | 8.78 | 114 |
| | MO | 58.6 | 58.4 | 5.77 | 9.85 | 66.1 | 72.8 | 6.70 | 110 |
| | RH | 59.7 | 59.5 | 6.06 | 10.2 | 67.5 | 74.7 | 7.17 | 111 |
| | SF | 56.5 | 56.1 | 7.12 | 12.6 | 66.0 | 74.8 | 8.77 | 113 |
| NE US | CASTNET | 57.4 | 56.9 | 7.42 | 12.9 | 66.6 | 78.1 | 11.4 | 117 |
| | MO | 74.4 | 73.7 | 11.4 | 15.4 | 89.8 | 104 | 13.8 | 115 |
| | RH | 76.0 | 75.1 | 11.9 | 15.6 | 92.0 | 107 | 14.8 | 116 |
| | SF | 72.6 | 71.3 | 14.5 | 20.0 | 91.8 | 114 | 21.9 | 124 |
| NE US single grid cell | CASTNET | 59.7 | 59.3 | 11.1 | 18.6 | 73.9 | 86.9 | 13.0 | 118 |
| | MO | 84.9 | 85.4 | 12.8 | 15.1 | 101 | 115 | 13.7 | 114 |
| | RH | 86.1 | 86.2 | 13.2 | 15.3 | 103 | 117 | 14.4 | 114 |
| | SF | 99.6 | 97.3 | 25.6 | 25.7 | 133 | 171 | 38.2 | 129 |

**Table 2:** Summary Statistics for the Daily Maximum 8-Hour (DM8H) O$_3$ over the globe and over the indicated regions in the US. Additional regions can be found in Supplemental Table S3.





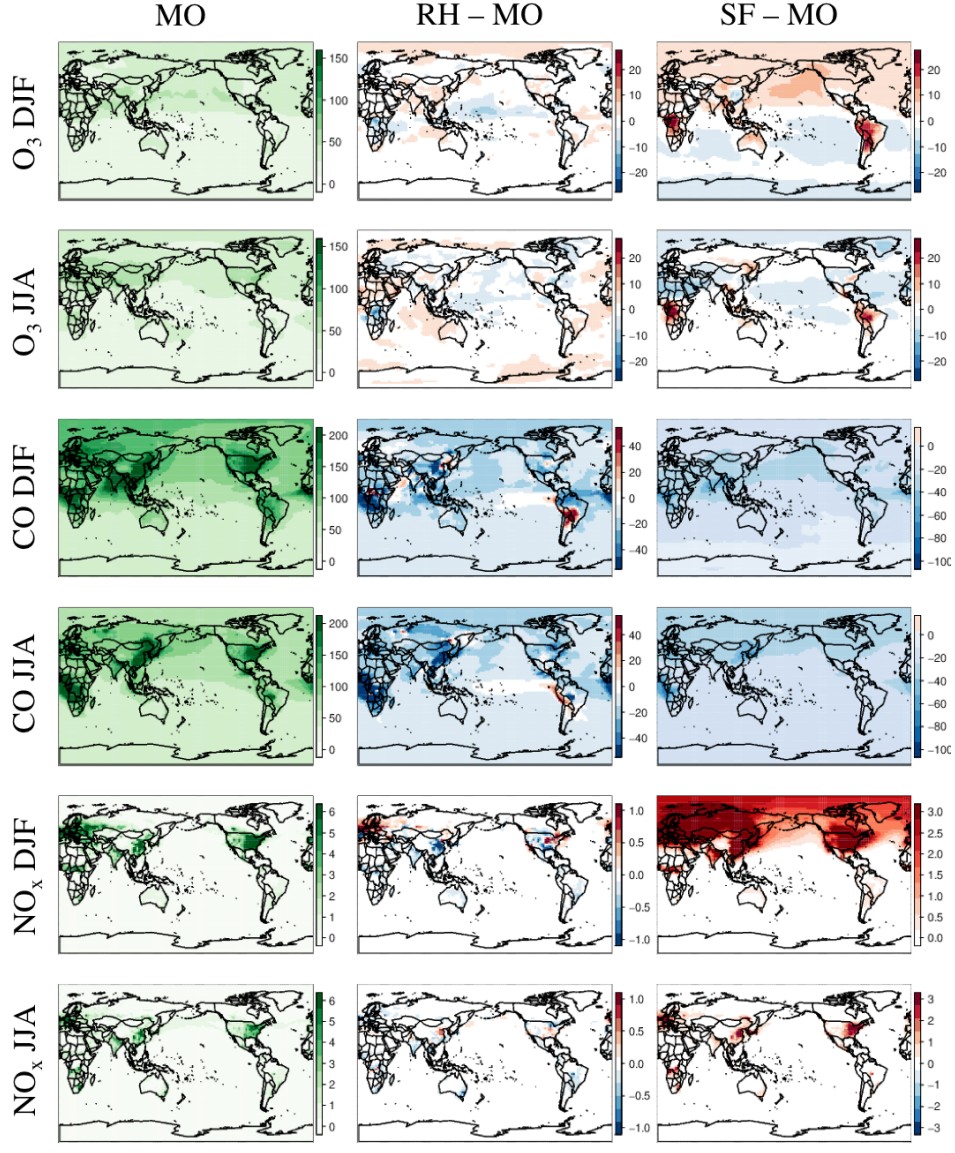

**Figure 1:** Maps of DJF and JJA $O_3$, CO, and $NO_x$ for MO, the difference between RH and MO, and between SF and MO for the year 2015. The chemical units are in ppb. Please note the difference in the chemical scales for each panel. Cool colors for the difference panels indicate MO is higher, and warm colors indicate that RH or SF is higher.





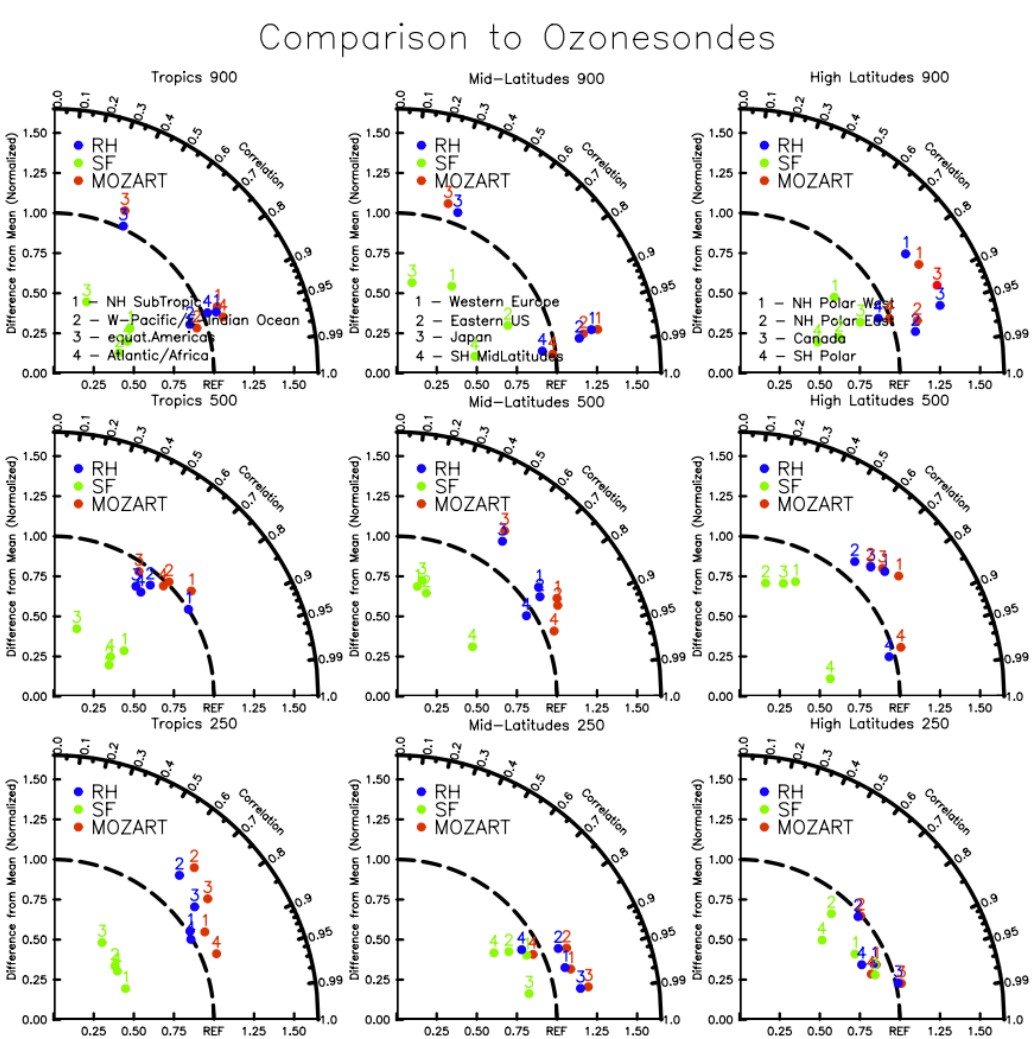

**Figure 2:** Taylor-like diagrams comparing the mean and correlation of the seasonal cycle between observations (present-day ozonesonde climatology (Tilmes et al., 2012) from 1995 to 2011, as in Figure 12 of Tilmes et al. (2015)) and simulations (red: MO, blue: RH, green: SF).





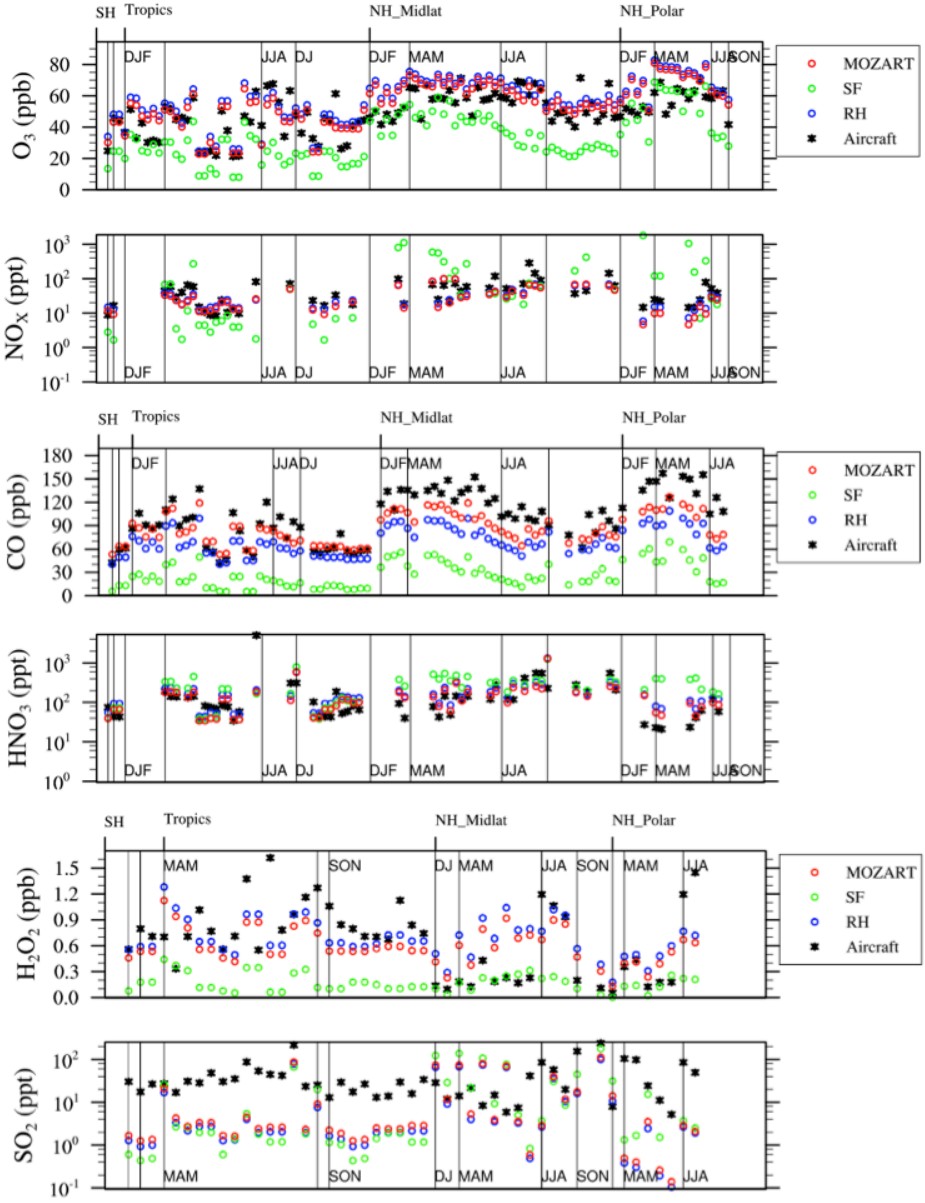

**Figure 3:** Relative differences between aircraft observations (black) and the MO, RH, and SF model configurations (colors) over different regions and seasons, averaged over $2 - 7$ km, for $O_3$, $NO_x$, $CO$, $HNO_3$, $H_2O_2$, and $SO_2$ as in Figure 17 of Tilmes et al. (2015).



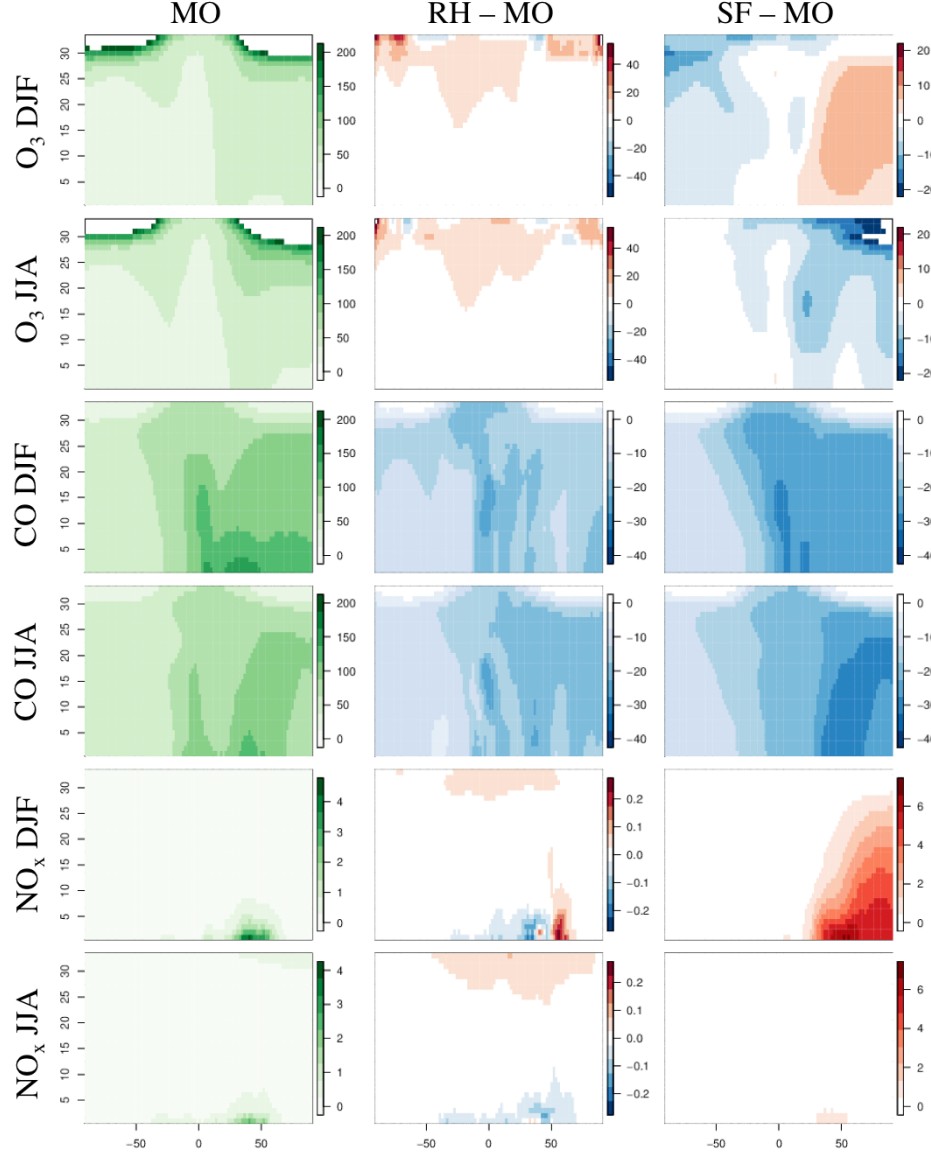

**Figure 4:** Zonal Plots of Seasonal $O_3$, CO, and $NO_x$ for MO, the difference between RH and MO, and between SF and MO for the year 2015. The vertical axis is the model level, and the chemical units are in ppb. Please note the different vertical axis in each row. Cool colors for the different panels indicate MO is higher, and warm colors indicate that RH or SF is higher.





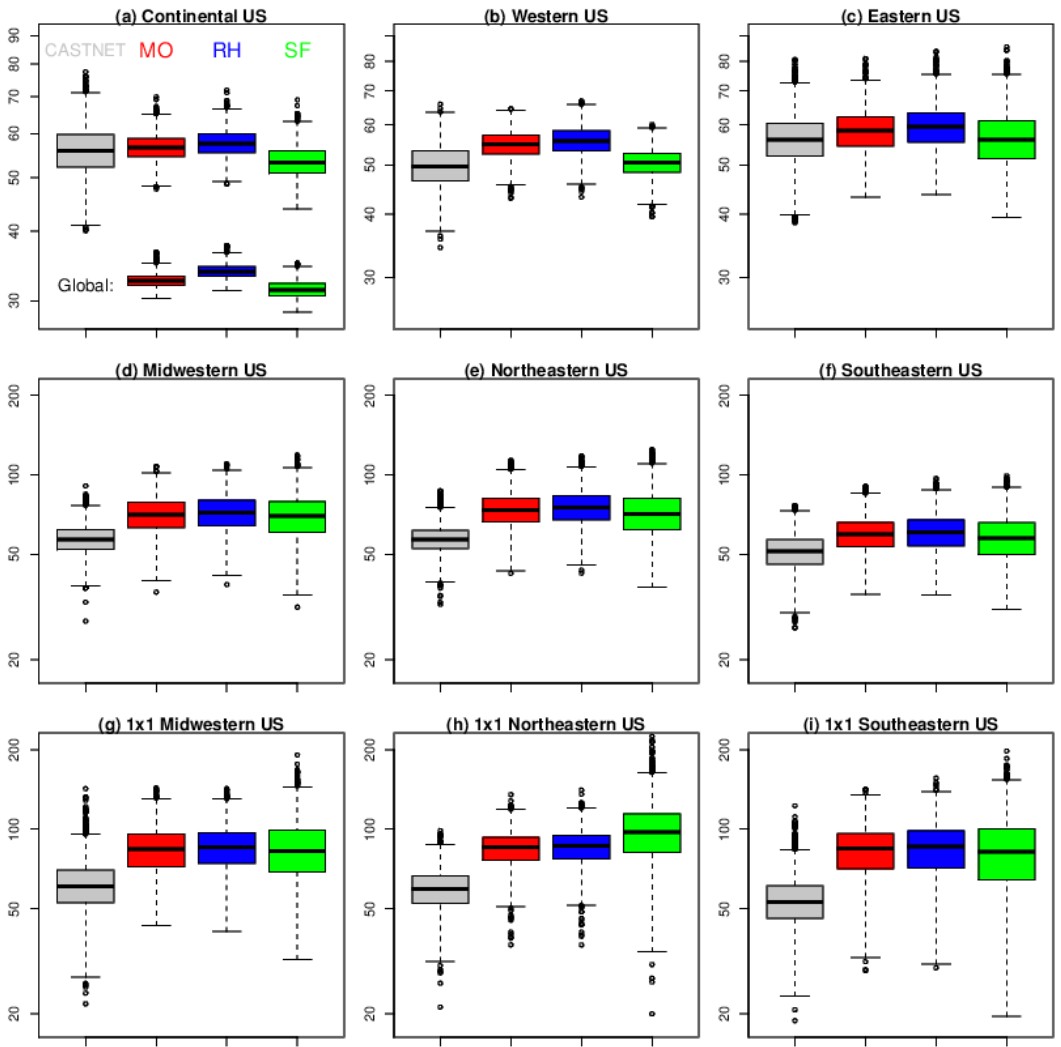

**Figure 5:** Surface JJA DM8H $O_3$ boxplots for the 1991 – 2014 data for CASTNET (grey), MO (red), RH (blue), and SF (green) averaged over the various regions. Global boxplots are included along with the Continental US. The units are in ppb, and for each boxplot the box contains the Inter Quartile Range (IQR), the horizontal line within the box is the median, and the whiskers extend out to the farthest point which is within 1.5 times the IQR with circles indicating any outliers. Note the scale difference between the top row and the rest of the panels.



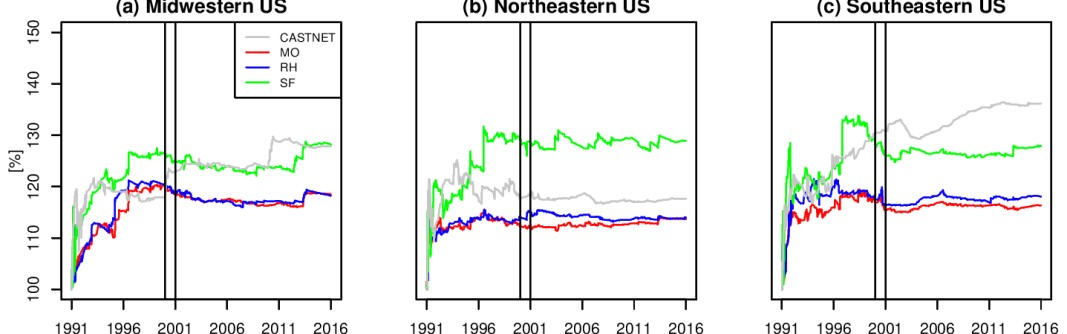

**Figure 6:** The relative difference (%) between the 99$^{th}$ percentile and the 90$^{th}$ percentile of JJA DM8H O$_3$ for CASTNET and the three mechanisms over three regions as a function of increasing length of simulation, from 1 day up to the full 25 years simulated. The vertical bars indicate the year 2000, for which the emissions for all three simulations were cycled.





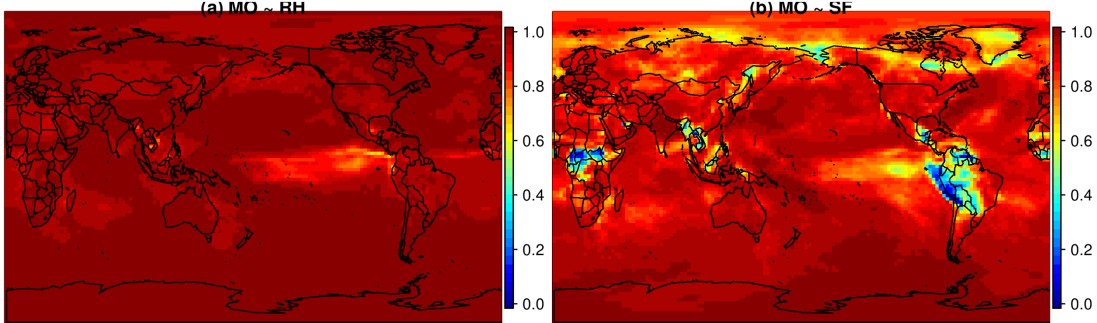

**Figure 7:** $R^2$ values calculated at every grid-cell (for the full 1990 – 2015 DM8H $O_3$ JJA time series) for MO and RH (left) and MO and SF (right).

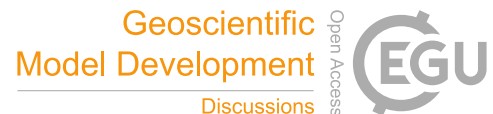



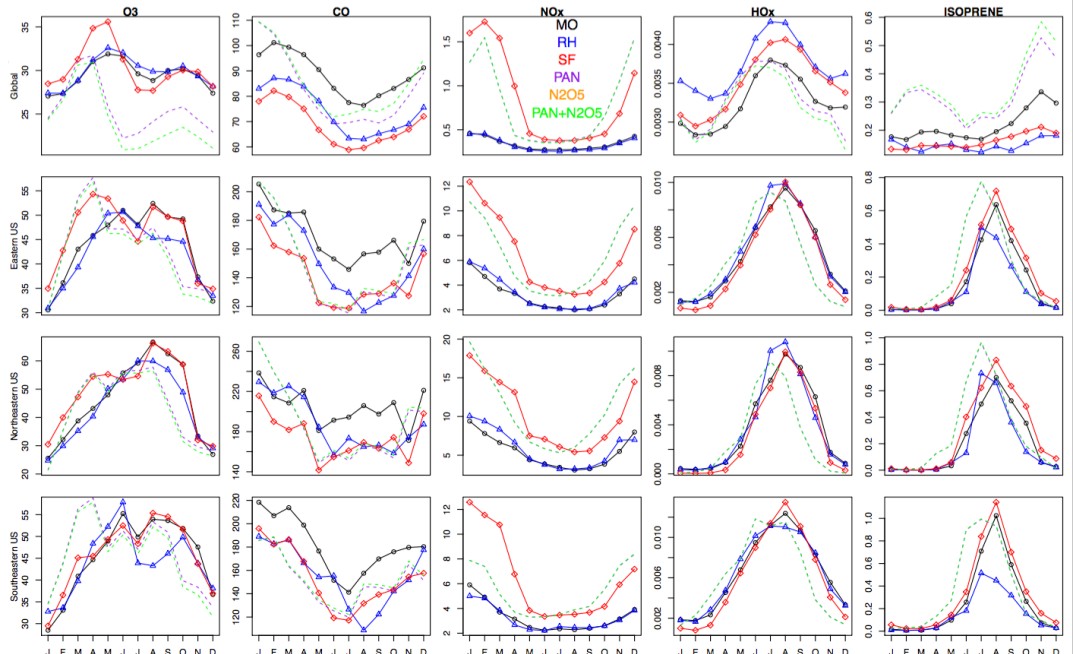

**Figure 8:** Seasonal time series for $O_3$, CO, $NO_x$, $HO_x$, and ISOP for MO (black), RH (blue), and SF (red) for a single year, averaged over different regions. The units are in ppb. Note the different scales in each panel. Also included are three sensitivity tests conducted with the SF mechanism: adding PAN chemistry (purple), adding $N_2O_5$ chemistry (orange), and adding both PAN and $N_2O_5$ chemistry (green).





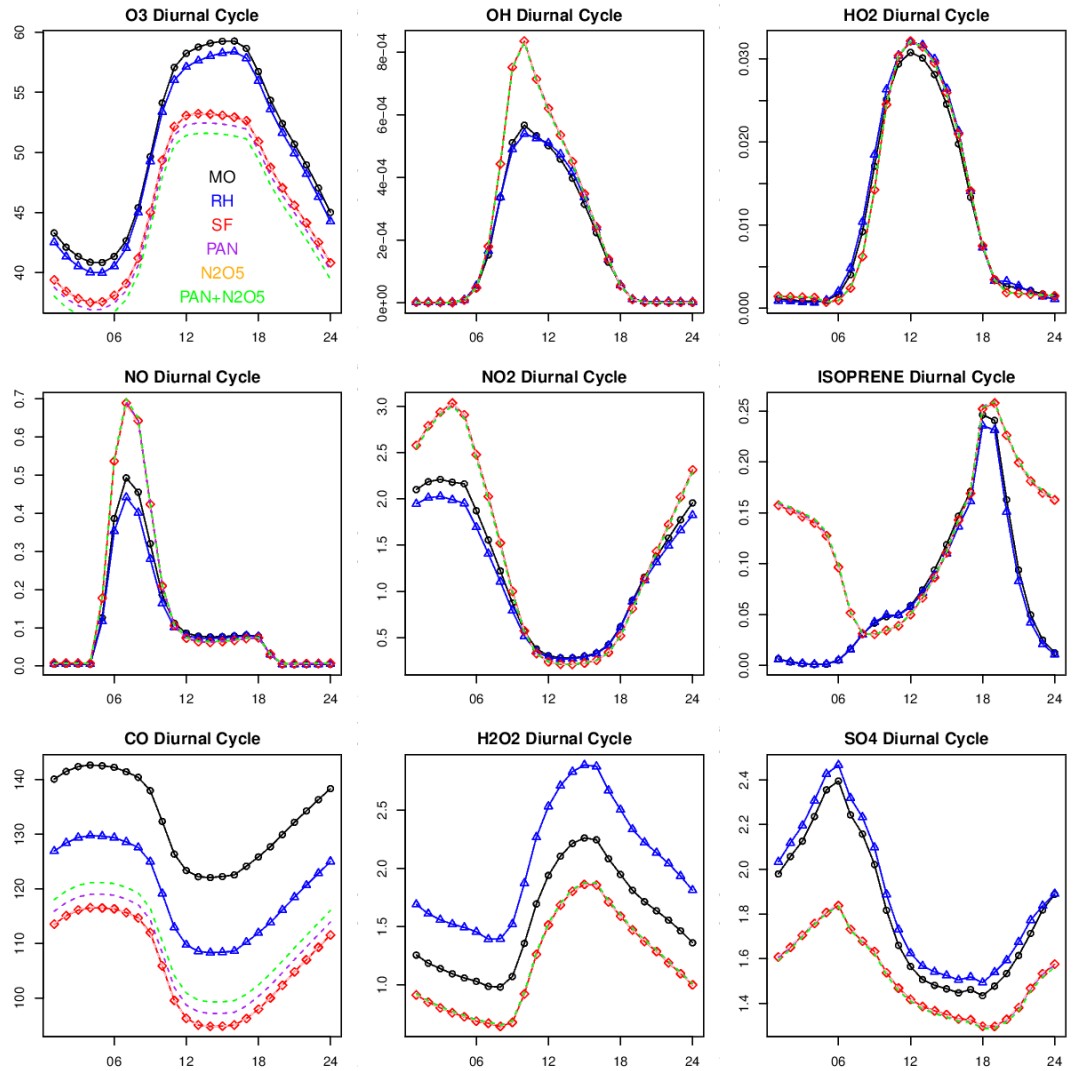

**Figure 9:** Example diurnal time series for various species for MO (black circles), RH (blue triangles), and SF (red diamonds) averaged over a single grid cell in the central US. The units are in ppb. Also included are three sensitivity tests conducted with the SF mechanism: adding PAN chemistry (purple), adding $N_2O_5$ chemistry (orange), and adding both PAN and $N_2O_5$ chemistry (green).





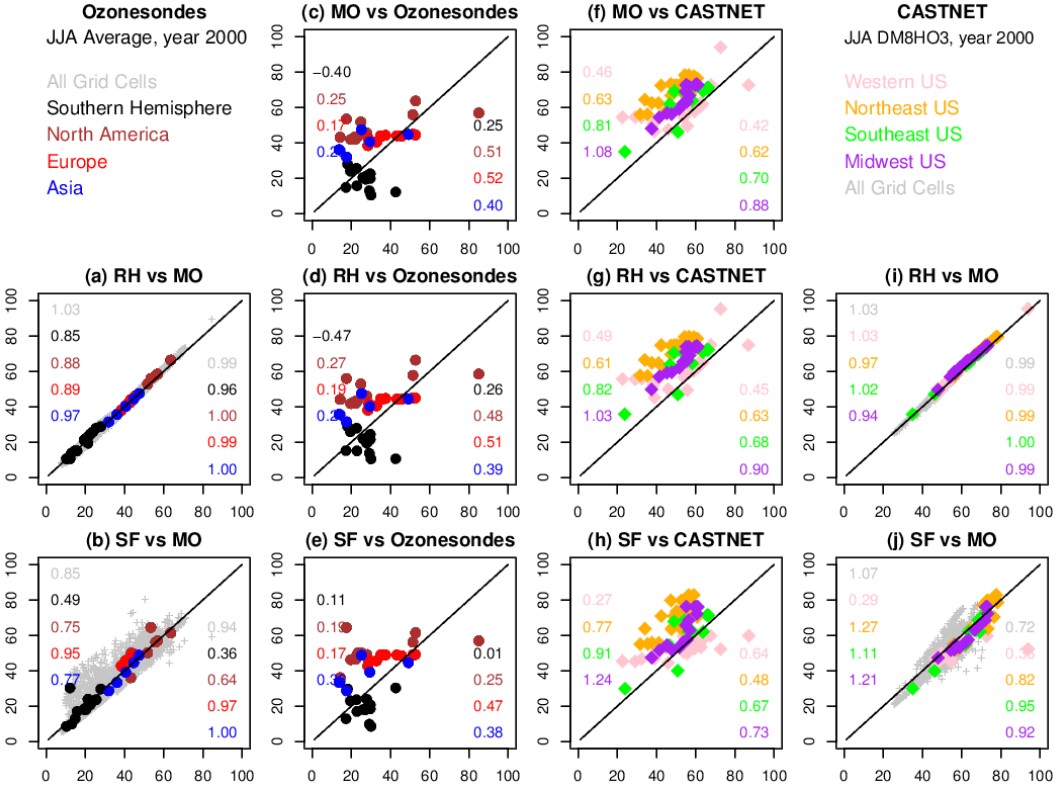

**Figure 10:** Scatterplots comparing model results to observations (two center columns) and to each other (two outer columns). Global regions (left) compare model results to ozonesondes (JJA averages), while regions within the US (right) compare the model results to CASTNET surface observations (JJA DM8H $O_3$). For the model-to-model comparisons, grey symbols additionally compare every grid cell in the model output. The numbers indicate the slope (upper left) and $R^2$ values (right) for each region. Each panel is labeled with the following convention: "y-axis" vs "x-axis."