# Peer review of "Evaluating Simplified Chemical Mechanisms within CESM Version 1.2 CAM-chem (CAM4): MOZART-4 vs. Reduced Hydrocarbon vs. Super-Fast Chemistry"

_Geoscientific Model Development, 2018_

## Editor Comment (EC1) · F. O'Connor (Editor) · 20 Feb 2018

Dear Benjamin and co-authors,

As the topical editor for your manuscript, may I remind you that I'm requesting that the data be made available on a suitable repository with a digital object identifier (DOI) before final publication of your manuscript in GMD. Reference to this repository can then be included in your final manuscript. I trust that this will be acceptable to you.

Regards, Fiona O'Connor

---

## Referee Comment (RC1) · Anonymous Referee #1 · 13 Mar 2018

Brown-Steiner et al present a study of how two reduced chemical mechanisms perform in a number of comparisons against the more comprehensive MOZART scheme (MO). The Reduced Hydrocarbon mechanism (RH) contains around half the number of MOZART reactions, whilst the Super-Fast mechanism (SF) is about 1/6 of MO. The work has been done to explore how much of a compromise it is by choosing one of these simple mechanisms over the more comprehensive scheme, when considering computational time gains versus accuracy of the chemical predictions.

The model runs have been conducted on a global scale for 25 years. This has enabled

the authors to pull out modelled data to compare with each other, and for the time periods and locations of ozonesonde, aircraft and CASTNET observations. Given how small SF is, it performs unexpectedly well, particularly against the CASTNET data, and in some cases better than the other two schemes. The RH scheme often tracks MO quite well, but with some exceptions, particularly for CO.

The Super-Fast mechanism could be used to explore chemical sensitivity studies in a fraction of the time it would take to run MOZART in locations of low biogenic activity.

I think the manuscript fits within the GMD journal remit and should be published. I have a few minor comments and queries.

General comments:

Please check throughout for the consistency about the length of the run. Page 6 line 1. 'we use MERRA.. for 25 years (1990 – 2014)'. However figure 1 plots maps for the year 2015, and figure 4 shows vertical distributions for 2015? Figure 6 also looks like it starts at 1991, not 1990 and carries on beyond 2014. Same applies to the statement on page 9, end of line 13 about the run being 1990-2014. Also figure 7 'for the full 1990-2015 time series', which is 26 years.

Section 2.2.1 MOZART-4. This section is very short and doesn't give the reader much information about MOZART other than to go searching through the suggested literature. I think a bit more information on what the scheme includes (e.g. how many alkanes/aromatics/biogenic species are considered explicitly) and omits would be useful, particularly as it is being used as the benchmark scheme.

Page 9 line 1. Please describe what 'BAM' means

Page 10. Line 20. There is a single line describing figure 6 and I then didn't fully understand the results drawn from it. My assumption is that the range of ozone at each time step in the model has been extracted for the region and the differences in the percentiles plotted here (although the figure caption says the CASTNET observations

are only for JJA?). The results say it takes 5-10 years for the models to stabilise, but the plots also show that the CASTNET observations themselves need 5-10 years to stabilise? If the models are behaving similarly to the observations, why would we need the spin up? There's a bit of a leap of understanding, so I think a bit more description is needed. I can see why you would expect the range in predicted ozone to settle as times goes on using constant emissions – but why also in the observations?

Page 14 line 16 spelling. 'mechanism', not 'mechanisms'.

Figures:

Figure 5, panels g,h and i. what does the 1x1 refer to?

Figure 7. Titles overlap with plots.

Figure 8. I'm struggling to see the orange N2O5 line in any of these plots. It could be that the line is hidden under the PAN+N2O5 line, but given the variation between PAN+N2O5 and N2O5 in the global ozone plot, I expected to see it?

Figure 8. Which single year are these seasonal cycles for? Why was this particular year chosen? How much variability is there between the first year run (1990) and the last (2014)?

Figure 9. Please give the location of the grid cell, (lon, lat).

Figure 10. please add units.

Figures, general comment:

About half way through the figures the colour scheme changes. In figures 2,3,5 and 6 MO is red, RH blue and SF is green. Later in figures 8 and 9, MO is black, RH is still blue but SF is now red. For the quick skimming reader, the assumption is that red is the benchmark scheme. It's a bit confusing.

[Figure]

2018.

GMDD

Interactive
comment

---

## Referee Comment (RC2) · Anonymous Referee #2 · 21 Mar 2018

Brown-Steiner et al. have performed an evaluation of the performance of a model of atmospheric chemistry run with three different chemical mechanisms to understand how big an impact there is by choosing a different chemical mechanism (network). This is an interesting paper, an important bit of science, and one of only a few examples in the literature to do this sort of work in a 3D sense. Evaluating the performance of these different mechanisms enables sound conclusions to be drawn about their utility. The aim is to see if a very simple mechanism, which would enable much longer (or many more) integrations as solving the coupled ODEs in the chemical network is computa-

tionally very expensive, is suitable. This is a very well written paper and I could hardly spot any typos or grammatical errors above those spotted by reviewer #1. However, I would like to see further experiments performed before I would recommend that this be published. As it stands, I don't think the suitability of the Super Fast (SF – simple chemistry) scheme to be used beyond a present day set up has been demonstrated. And I think this is key for the argument that the SF scheme is suitable.

The present work focuses on fairly long integrations (25 years or so), where anthropogenic emissions are fixed but interactive biogenic emissions can change as the meteorology in the model changes. Some very nice analysis is then performed against surface observations of O3 which emphasises that for these conditions, the SF scheme performs well – in accord with the other more complex schemes.

But, to be convinced that the SF scheme is suitable for long simulations of transient forcing, I would like to see simulations that test the response of the chemical schemes to the sorts of changes that have happened over the Anthropocene and for which the SF scheme may well end up being used for within CEMS (i.e. CMIP/AerChemMIP type experiments). For example, I would like to see, as a minimum, a set of simulations using ACCMIP pre-industrial emissions (you could keep the meteorology fixed as it is if that makes things easier) so that we can see what happens between these different schemes when they are perturbed with significant changes in NOx and VOC (CH4).

I also am a bit concerned with the reproducibility of these experiments outside of the team working on this. There are no mentions of compsets (is that the word used within the CESM model set ups? I'm going from here http://www.cesm.ucar.edu/models/cesm1.2/casename_conventions_cesm.html) that have been used. Citing some rather old papers as the sources of the rate constants and reactions used, for example in the Reduced Hydrocarbon scheme, makes it difficult for others to test the schemes without large potential for making translational errors (I know, I have made many myself!). I would propose that the mechanism data files be made available (perhaps in a simple scv format?) or at least the compsets for

these experiments be made available so that others can perform their own tests. GMD is a journal dedicated to holding high standards with code and I think that the mechanisms should be treated as a bit of complex code that should be archived in order to be more easily tested. This would be desirable but I can appreciate that this may not be top priority.

Minor corrections/comments:

Page 8, line 23: A key conclusion of Squire et al. was that sign of the response to changes in emissions of isoprene was different in SF compared to more complex schemes traceable to our best understanding of the chemistry of isoprene (ie. the MCM). I think this needs to be acknowledged here in addition to current acknowledgment that "there are biases in regions of high biogenic chemistry".

I note from Table S2 and from the discussion in Squire et al., that the SF scheme does not include NO3. Presumably the bias in isoprene at night (Figure 9) could be solved by simulating NO3 in the SF scheme? Have the authors considered this? It was not clear from the manuscript if that was tested in addition to the nice tests looking at the impacts of adding in PAN and N2O5.

Table S2, reaction 14: "idential" should be "identical".

Figure 8: Axis labels are way too small. Please make bigger. As above for Figure 9.

How are the VOC emissions dealt with between the different schemes? I presume that there are different amounts of VOC that go into the simulations? Please can you clarify the magnitude and distribution amongst molecules of the VOC emissions. Emissions are a key part of the chemical mechanism in my opinion.

---

## Author Response (AR1)

Author response to reviews for "Evaluating Simplified Chemical Mechanisms within Present-Day Simulations of CESM Version 1.2 CAM-chem (CAM4): MOZART-4 vs. Reduced Hydrocarbon vs. Super-Fast Chemistry"
By Benjamin Brown-Steiner et al.

We first want thank the reviewers for their valuable and insightful comments, and for taking the time to review our manuscript.

We now respond to the reviewer comments, which are reproduced in black text below. Our responses follow immediately in red text, and any additions to the manuscript are included in italic red text, along with Line references, which refer to their locations in the revised manuscript.
* * *
Editor, F. O'Connor:

Dear Benjamin and co-authors,
As the topical editor for your manuscript, may I remind you that I'm requesting that the data be made available on a suitable repository with a digital object identifier (DOI) before final publication of your manuscript in GMD. Reference to this repository can then be included in your final manuscript. I trust that this will be acceptable to you.
Regards, Fiona O'Connor

We have uploaded the relevant data to a repository hosted on the MIT domain (http://dspace.mit.edu/handle/1721.1/114993). The Data/Code availability sections have been updated to direct readers to this repository, as is discussed in the comments below.
* * *
Anonymous Referee #1

Brown-Steiner et al present a study of how two reduced chemical mechanisms perform in a number of comparisons against the more comprehensive MOZART scheme (MO). The Reduced Hydrocarbon mechanism (RH) contains around half the number of MOZART reactions, whilst the Super-Fast mechanism (SF) is about 1/6 of MO. The work has been done to explore how much of a compromise it is by choosing one of these simple mechanisms over the more comprehensive scheme, when considering computational time gains versus accuracy of the chemical predictions.

The model runs have been conducted on a global scale for 25 years. This has enabled the authors to pull out modelled data to compare with each other, and for the time periods and locations of ozonesonde, aircraft and CASTNET observations. Given how small SF is, it performs unexpectedly well, particularly against the CASTNET data, and in some cases better than the

other two schemes. The RH scheme often tracks MO quite well, but with some exceptions, particularly for CO.

The Super-Fast mechanism could be used to explore chemical sensitivity studies in a fraction of the time it would take to run MOZART in locations of low biogenic activity.

I think the manuscript fits within the GMD journal remit and should be published. I have a few minor comments and queries.

General comments:

Please check throughout for the consistency about the length of the run. Page 6 line 1. 'we use MERRA.. for 25 years (1990 – 2014)'. However figure 1 plots maps for the year 2015, and figure 4 shows vertical distributions for 2015? Figure 6 also looks like it starts at 1991, not 1990 and carries on beyond 2014. Same applies to the statement on page 9, end of line 13 about the run being 1990-2014. Also figure 7 'for the full 1990-2015 time series', which is 26 years.

We ran the simulations for 26 years (1990 – 2015) and used the first year as spin up, so the analyses are for 25 years (1991 – 2015). We have corrected the descriptions and added a line indicating the 1990 year as spin up:

Page 6, Line 2: "…for 2*6* years (1990 – 201*5*)…"
Page 6, Lines 3-4: "*The year 1990 is dropped to allow for spin-up.*"
Page 9, Line 24-25: "…sites throughout the *1991 – 2015* period…"
Figure 7, Caption: "…the full *1991* – 2015…"

Section 2.2.1 MOZART-4. This section is very short and doesn't give the reader much information about MOZART other than to go searching through the suggested literature. I think a bit more information on what the scheme includes (e.g. how many alkanes/aromatics/biogenic species are considered explicitly) and omits would be useful, particularly as it is being used as the benchmark scheme.

We have added additional details and point again to Emmons et al. (2010) for a complete description.

Section 2.2.1: "*As described in detail in Emmons et al. (2010), MOZART-4 mechanism is a tropospheric mechanism that contains 85 gas-phase species and 12 bulk aerosol species, with 39 photolysis and 157 gas-phase reactions. Large alkanes, alkenes, and aromatics are lumped together (BIGALK, BIGENE, and TOLUENE, respectively), and monoterpenes are lumped together as C10H16 and treated as α-pinene.*"

Page 9 line 1. Please describe what 'BAM' means

Page 9, Line 10: "…the *Bulk Aerosol Model* (BAM) *(see Tilmes et al., 2015)*…"

Page 10. Line 20. There is a single line describing figure 6 and I then didn't fully understand the results drawn from it. My assumption is that the range of ozone at each time step in the model has been extracted for the region and the differences in the percentiles plotted here (although the

figure caption says the CASTNET observations are only for JJA?). The results say it takes 5-10 years for the models to stabilise, but the plots also show that the CASTNET observations themselves need 5-10 years to stabilize? If the models are behaving similarly to the observations, why would we need the spin up? There's a bit of a leap of understanding, so I think a bit more description is needed. I can see why you would expect the range in predicted ozone to settle as times goes on using constant emissions – but why also in the observations?

In a concurrent paper under review in ACPD (Brown-Steiner et al., in review) we expand on the implications of this figure. We add additional clarifications of Figure 6 and point to reader to the concurrent paper for additional discussion. We also add this paper to the references.

Page 11, Line 5: "*Brown-Steiner et al. (in review, ACPD) examines these implications, and also concludes that it takes approximately 10-years for long-term signals to emerge from meteorological variability. These results demonstrate the challenge in examining chemical signals in highly variable data, particularly if there are trends or changes to the ozone distribution, as is seen in the CASTNET data for the Southeastern US.*"

Page 14 line 16 spelling. 'mechanism', not 'mechanisms'.

Corrected.

Figures:

Figure 5, panels g,h and i. what does the 1x1 refer to?

They are individual grid cells within each region. This has been added to the caption.

Figure 5, caption: "*Plots g, h, and i are individual grid cells from within each region*."

Figure 7. Titles overlap with plots.

The figure has been updated to correct this.

Figure 8. I'm struggling to see the orange N2O5 line in any of these plots. It could be that the line is hidden under the PAN+N2O5 line, but given the variation between PAN+N2O5 and N2O5 in the global ozone plot, I expected to see it?

The figure has been updated so that each line is more easily discernable.

Figure 8. Which single year are these seasonal cycles for? Why was this particular year chosen? How much variability is there between the first year run (1990) and the last (2014)?

Year 2015 was selected as a representative year, and we find some variability year-to-year due to meteorology, but all models tend to demonstrate the same year-to-year variability. For the sensitivity tests with PAN and N2O5, we only ran 2 years and selected the 2nd year (1991). We

also discovered a bug in the plotting code in the sensitivity tests which has been corrected. Some of the seasonal cycles were offset by 2 months, and but does not impact our conclusions.

Figure 8, Caption: "…single year *(2015)*, averaged…"

Figure 9, Caption: "…SF mechanism *(which were ran only for 2 years, 1990 – 1991, with 1991 being plotted here)*…"

Figure 9. Please give the location of the grid cell, (lon, lat).

This has been added to the caption of Figure 9.

Figure 9, Caption: "*(100˚ west and 47˚ north)*"

Figure 10. please add units.

Added.

Figures, general comment:
About half way through the figures the colour scheme changes. In figures 2,3,5 and 6 MO is red, RH blue and SF is green. Later in figures 8 and 9, MO is black, RH is still blue but SF is now red. For the quick skimming reader, the assumption is that red is the benchmark scheme. It's a bit confusing.

All figures have been updated to remain consistent with the color schemes (MO red, RH blue, SF green).
* * *
Anonymous Referee #2

Brown-Steiner et al. have performed an evaluation of the performance of a model of atmospheric chemistry run with three different chemical mechanisms to understand how big an impact there is by choosing a different chemical mechanism (network). This is an interesting paper, an important bit of science, and one of only a few examples in the literature to do this sort of work in a 3D sense. Evaluating the performance of these different mechanisms enables sound conclusions to be drawn about their utility. The aim is to see if a very simple mechanism, which would enable much longer (or many more) integrations as solving the coupled ODEs in the chemical network is computationally very expensive, is suitable. This is a very well written paper and I could hardly spot any typos or grammatical errors above those spotted by reviewer #1. However, I would like to see further experiments performed before I would recommend that this be published. As it stands, I don't think the suitability of the Super Fast (SF – simple chemistry) scheme to be used beyond a present day set up has been demonstrated.

And I think this is key for the argument that the SF scheme is suitable. The present work focuses on fairly long integrations (25 years or so), where anthropogenic emissions are fixed but

interactive biogenic emissions can change as the meteorology in the model changes. Some very nice analysis is then performed against surface observations of O3 which emphasises that for these conditions, the SF scheme performs well – in accord with the other more complex schemes.

But, to be convinced that the SF scheme is suitable for long simulations of transient forcing, I would like to see simulations that test the response of the chemical schemes to the sorts of changes that have happened over the Anthropocene and for which the SF scheme may well end up being used for within CEMS (i.e. CMIP/AerChemMIP type experiments). For example, I would like to see, as a minimum, a set of simulations using ACCMIP pre-industrial emissions (you could keep the meteorology fixed as it is if that makes things easier) so that we can see what happens between these different schemes when they are perturbed with significant changes in NOx and VOC (CH4).

The Super-Fast mechanism was included in ACCMIP studies in both historical and future conditions (as reviewed in the Supplemental Material), although a full description is not in the scientific literature. We hope this manuscript allows other researchers to more easily find and utilize the SF mechanism and that this manuscript can serve as a baseline for future simulations and testing.

As this study is intended as a demonstration of the Super-Fast mechanism's utility, rather than a comprehensive evaluation, and as we have only analyzed present-day, we have added "Present-Day" to the manuscript title "Evaluating Simplified Chemical Mechanisms within _Present-Day_ Simulations of CESM …"

Demonstrating the capabilities of the Super-Fast mechanism for long-term transient forcing was beyond our scope and capabilities, and we agree that more work needs to be done with the Super-Fast mechanism (as well as the Reduced Hydrocarbon mechanism), and that care should always be taken when any model component is utilized outside of previously demonstrated periods and conditions. As such, we have made sure that the mechanism files are available (see reproducibility comment below), and we are in discussion as how to best integrate these mechanisms into available forms at the NCAR/CESM website.

In light of this, we have added language that makes clear the limitations of this manuscript and future research directions which will be needed to further study the utility and capabilities of the Super-Fast and Reduced Hydrocarbon mechanisms, as well as made clear where the code can be accessed (see comment below).

Abstract: "Here we present and compare three 25-year _present-day_ offline simulations…"

Page 14, Lines 27-28: "…we have compared three chemical mechanisms of different levels of complexity within the CESM CAM-chem framework _for present-day chemical and climatological conditions._"

We have also added language in the conclusions highlighting that this study is only for present-day conditions:

Page 14, Lines 36-40: "*We examine present-day chemistry with MO, RH, and SF. Both MO and SF have been compared in other model intercomparisons, including for preindustrial conditions (see the Supplemental Material for additional information). We hope that the analysis presented in this paper, and the availability of the mechanism files (Supplemental Material) will provide a baseline for continuing research of both the RH and SF mechanisms.*"

I also am a bit concerned with the reproducibility of these experiments outside of the team working on this. There are no mentions of compsets (is that the word used within the CESM model set ups? I'm going from here http://www.cesm.ucar.edu/models/cesm1.2/casename_conventions_cesm.html) that have been used. Citing some rather old papers as the sources of the rate constants and reactions used, for example in the Reduced Hydrocarbon scheme, makes it difficult for others to test the schemes without large potential for making translational errors (I know, I have made many myself!). I would propose that the mechanism data files be made available (perhaps in a simple scv format?) or at least the compsets for these experiments be made available so that others can perform their own tests. GMD is a journal dedicated to holding high standards with code and I think that the mechanisms should be treated as a bit of complex code that should be archived in order to be more easily tested. This would be desirable but I can appreciate that this may not be top priority.

We have added to the data uploaded to the archive (http://dspace.mit.edu/handle/1721.1/114993) the chemical mechanism input files (reduced_hydrocarbon.in and superfast.in) and add additional text to the methods section and code availability section to direct readers to various CESM/NCAR resources. We clarify that we use the FMOZSOA compset for the MO simulation and make modifications to the chemical mechanism input file and speciation of species, as described in the text.

Page 6, Lines 22-25: "*The chemical mechanism input files for MO is available in the standard CESM release (http://www.cesm.ucar.edu/models/cesm1.2/) and the chemical mechanism input files used for RH and SF are archived (see section on Code Availability)*"

Page 6, Lines 36-39: "*We use the FMOZSOA compset (see http://www.cesm.ucar.edu/models/cesm1.2/cesm/doc/modelnl/compsets.html) and make modifications to the chemical mechanism input files (see section on Coda Availability) and emission files for the following mechanisms.*"

Code Availability: "*The chemical mechanism files for both RH (reduced_hydrocarbon.in) and SF (superfast.in) are included in the Supplemental Material.*"

Supplemental Material: "*The SF mechanism is in the CESM code archive as an unsupported chemical mechanism, which can be activated using the option '-chem super_fast_llnl'.*"

Minor corrections/comments:

Page 8, line 23: A key conclusion of Squire et al. was that sign of the response to changes in emissions of isoprene was different in SF compared to more complex schemes traceable to our best understanding of the chemistry of isoprene (ie. The MCM). I think this needs to be acknowledged here in addition to current acknowledgement that "there are biases in regions of high biogenic chemistry".

We have added language to this section to highlight this Squire et al. (2015) conclusion:

Page 8, Lines 37-40: "*Schnell et al. (2015) also conclude that the SF mechanism responds differently than other more complex mechanisms, particular under different $O_x$ production regimes (e.g. SF shows a net increase in $O_x$ production when isoprene emissions increase in $NO_x$-limited regions, which the other mechanisms show a net decrease, or little change).*"

I note from Table S2 and from the discussion in Squire et al., that the SF scheme does not include NO3. Presumably the bias in isoprene at night (Figure 9) could be solved by simulating NO3 in the SF scheme? Have the authors considered this? It was not clear from the manuscript if that was tested in addition to the nice tests looking at the impacts of adding in PAN and N2O5.

The authors did not test the addition of NO3 to the Super-Fast scheme. The addition of the PAN and N2O5 sensitivity tests are intended primarily as a demonstration of the type of simulations and sensitivity studies that the Super-Fast mechanism allows for.

To this point, we have added language in the manuscript that speculates about the addition of NO3 to the Super-Fast mechanism:

Page 15, Lines 16-18: "*The SF mechanism does not include $NO_3$, which may also explain some of the nighttime biases. Future simulations in which $NO_3$ chemistry is added to the SF mechanism may correct some of these biases.*"

Table S2, reaction 14: "idential" should be "identical".

Corrected.

Figure 8: Axis labels are way too small. Please make bigger. As above for Figure 9.

Figure 8 font sizes have been increased, and we will work with the editors to see if this plot can be included as a full page. If not, we will work with the editors to make sure they are readable. Figure 9 font sizes have been increased.

How are the VOC emissions dealt with between the different schemes? I presume that there are different amounts of VOC that go into the simulations? Please can you clarify the magnitude and distribution amongst molecules of the VOC emissions. Emissions are a key part of the chemical mechanism in my opinion.

Supplemental Table S1 includes the mapping of VOC species from MO to RH (which is discussed in section 2.2.2). For SF, we mapped only the MO ISOP species directly to the SF ISOP species. We have also added some text discussing this point:

[revised manuscript text omitted]

**Gas-Phase Reactions**

| | Reactants | | Products | Rate | Relation to Emmons et al. (2010) |
|---|---|---|---|---|---|
| (1) | $O_3 + OH$ | $\rightarrow$ | $HO_2 + O_2$ | $1.70E\text{-}12\cdot exp(-940/T)$ | identical |
| (2) | $HO_2 + O_3$ | $\rightarrow$ | $2\cdot O_2 + OH$ | $1.00E\text{-}14\cdot exp(-490/T)$ | identical |
| (3) | $HO_2 + OH$ | $\rightarrow$ | $H_2O + O_2$ | $4.80E\text{-}11\cdot exp(250/T)$ | identical |
| (4) | $HO_2 + HO_2$ | $\rightarrow$ | $H_2O_2 + O_2$ | $(2.3E\text{-}13\cdot exp(600/T)+1.7E\text{-}33\cdot[M]\cdot exp(1000/T))\cdot (1 + 1.4E\text{-}21\cdot[H2O]\cdot exp(2200/T))$ | identical |
| (5) | $H_2O_2 + OH$ | $\rightarrow$ | $H_2O + HO_2$ | $1.80E\text{-}12$ | identical |
| (6) | $NO + O_3$ | $\rightarrow$ | $NO_2 + O_2$ | $3.00E\text{-}12\cdot exp(-1500/T)$ | identical |
| (7) | $HO_2 + NO$ | $\rightarrow$ | $NO_2 + OH$ | $3.50E\text{-}12\cdot exp(250/T)$ | different rates |
| (8) | $NO_2 + OH + M$ | $\rightarrow$ | $HNO_3$ | $ko=1.80E\text{-}30\cdot(300/T)^{3.00}; ki=2.80E\text{-}11; f=0.60$ | identical |
| (9) | $CH_4 + OH$ | $\rightarrow$ | $CH_3O_2 + H_2O$ | $2.45E\text{-}12\cdot exp(-1775/T)$ | identical |
| (10) | $CO + OH$ | $\rightarrow$ | $HO_2$ | $ko\_m/(1+(ko\_m/k_\infty))\cdot 0.6\cdot exp((1/(1+log(ko\_m/k_\infty))^2)))+(k0/(1+(k0/k_{\infty\_}m)))\cdot 0.6\cdot exp(( (1/(1+(log(k0/k_{\infty\_}m))^2)))$ | simplified: no $CO_2$, $H \rightarrow HO_2$, see note A |
| (11) | $CH_2O + OH$ | $\rightarrow$ | $CO + H_2O + HO_2$ | $5.50E\text{-}12\cdot exp(125/T)$ | rates identical, simplified: $H \rightarrow HO_2$ |
| (12) | $CH_3O_2 + HO_2$ | $\rightarrow$ | $CH_3OOH + O_2$ | $4.10E\text{-}13\cdot exp(750/T)$ | identical |
| (13a) | $CH_3OOH + OH$ | $\rightarrow$ | $CH_3O_2 + H_2O$ | $2.70E\text{-}12\cdot exp(200/T)$ | in combination, equivalent |
| (13b) | $CH_3OOH + OH$ | $\rightarrow$ | $CH_3O + H_2O + OH$ | $1.10E\text{-}12\cdot exp(200/T)$ | |
| (14) | $CH_3O_2 + NO$ | $\rightarrow$ | $CH_2O + HO_2 + NO_2$ | $2.80E\text{-}12\cdot exp(300/T)$ | identical |
| (15) | $CH_3O_2 + CH_3O_2$ | $\rightarrow$ | $2\cdot CH_2O + 0.80\cdot HO_2$ | $9.50E\text{-}14\cdot exp(390/T)$ | different rates, simplified: 1 reaction instead of 2 |
| (16) | $H_2O + NO_2$ | $\rightarrow$ | $0.50\cdot HNO_3$ | $4.00E\text{-}24$ | no equivalent in MOZART, see note B |
| (17a) | $DMS + OH$ | $\rightarrow$ | $SO_2$ | $1.100E\text{-}11\cdot exp(-240/T)$ | |
| (17b) | $DMS + OH$ | $\rightarrow$ | $0.75\cdot SO_2$ | $2.00E\text{-}10\cdot exp(5820\cdot[M])/ ((2.00E29/[O2]) + exp(6280\cdot[M]))$ | different, see note C |
| (18) | $OH + SO_2 + M$ | $\rightarrow$ | $SO_4$ | $ko=3.30E\text{-}31\cdot(300/T)^{4.30}; ki=1.60E\text{-}12; f=0.60$ | different, see note C |
| (19) | $H_2O_2 + SO_2$ | $\rightarrow$ | $SO_4$ | aqueous chemistry (see note D) | no equivalent, see note C |
| (20) | $O_3 + SO_2$ | $\rightarrow$ | $SO_4$ | aqueous chemistry (see note D) | no equivalent, see note C |
| (21a) | $ISOP + OH$ | $\rightarrow$ | $2\cdot CH_3O_2$ | $2.70E\text{-}11\cdot exp(390/T)$ | |
| (21b) | $ISOP + OH$ | $\rightarrow$ | $ISOP$ | $2.70E\text{-}11\cdot exp(390/T)$ | different, see note E |
| (21c) | $ISOP + OH$ | $\rightarrow$ | $ISOP + 0.5\cdot OH$ | $2.70E\text{-}11\cdot exp(390/T)$ | |
| (22) | $ISOP + O_3$ | $\rightarrow$ | $.87\cdot CH_2O + 1.86\cdot CH_3O_2 + 0.06\cdot HO_2 + 0.05\cdot CO$ | $5.59E\text{-}15\cdot exp(-1814/T)$ | different, see note E |

NOTES:

A: For rate: $ko = 5.90E\text{-}33\bullet(300/T)1.4$; $k_\infty = 1.10E\text{-}12\bullet(T/300)1.3$; $ko\_m = ko\bullet[M]$; $k0 = 1.50E\text{-}13\bullet(T/300)0.6$; $k_{\infty\_}m = (2.10E9\bullet(T/300)6.1)/[M]$

B: $HNO_3$ chemistry included only as reaction 8 and 16, with reaction 16 involving heterogeneous chemistry paramaterization

C: DMS chemistry limited only to reaction with OH (reaction 17), $SO_4$ production simplified to reactions 18 with OH and 19 and 20 with aqueous chemistry (with a fixed pH in the cloud droplets)

D: Rate equations are included within the aerosol routines adapted from the MOZART-4 mechanism implimentation within CAM-chem

E: Isoprene chemistry paramaterized from UCI for ISOP + OH and from LLNL-IMAPCT for ISOP + $O_3$, see text for full details

**Table S2:** Full description of Super-Fast chemical mechanism as compared to the MOZART-4 mechanism of Emmons et al. (2010). Reaction rates are written out if they are of the Arrenhius form, or otherwise formulated. If the reaction rates are of the Troe form, they list the ko and ki parameters, as in Emmons et al. (2010). The simplifications made in

the SF are noted by indicating what species is missing or modified when compared to Emmons et al. (2010). Chemical species are the same as in Emmons et al. (2010).

| | | [ppbv] Mean | [ppbv] Median | [ppbv] Standard Deviation | [%] Variability | [ppbv] 90th Percentile | [ppbv] 99th Percentile | [ppbv] 99th - 90th Percentile | [%] |
|---|---|---|---|---|---|---|---|---|---|
| **W US** | CASTNET | 49.9 | 49.6 | 4.82 | 9.66 | 56.4 | 61.3 | 4.87 | 109 |
| | MO | 54.9 | 54.9 | 3.37 | 6.14 | 59.4 | 62.4 | 2.99 | 105 |
| | RH | 55.9 | 55.8 | 3.58 | 6.41 | 60.6 | 64.0 | 3.47 | 106 |
| | SF | 50.5 | 50.5 | 3.22 | 6.38 | 54.7 | 57.8 | 3.15 | 106 |
| **MW US** | CASTNET | 57.3 | 57.0 | 7.57 | 13.2 | 67.1 | 76.9 | 9.77 | 115 |
| | MO | 71.0 | 70.7 | 11.03 | 15.5 | 85.5 | 96.0 | 10.6 | 112 |
| | RH | 72.6 | 72.1 | 11.26 | 15.5 | 87.4 | 98.8 | 11.4 | 113 |
| | SF | 70.6 | 69.8 | 13.44 | 19.0 | 88.7 | 104 | 15.1 | 117 |
| **MW US single grid cell** | CASTNET | 62.4 | 60.6 | 14.5 | 23.2 | 81.7 | 104 | 22.7 | 128 |
| | MO | 84.6 | 83.8 | 16.2 | 19.2 | 106 | 126 | 19.4 | 118 |
| | RH | 86.2 | 85.3 | 15.8 | 18.4 | 107 | 126 | 19.5 | 118 |
| | SF | 85.4 | 82.5 | 22.4 | 26.2 | 116 | 149 | 32.8 | 128 |
| **SE US** | CASTNET | 51.5 | 51.3 | 8.18 | 15.9 | 62.3 | 71.9 | 9.64 | 115 |
| | MO | 60.2 | 59.5 | 9.22 | 15.3 | 72.5 | 83.7 | 11.2 | 115 |
| | RH | 61.3 | 60.5 | 10.1 | 16.6 | 74.9 | 87.9 | 12.9 | 117 |
| | SF | 58.6 | 57.8 | 11.4 | 19.4 | 73.9 | 87.1 | 13.2 | 118 |
| **SE US single grid cell** | CASTNET | 53.8 | 53.1 | 12.8 | 23.8 | 68.7 | 93.5 | 24.8 | 136 |
| | MO | 83.4 | 84.7 | 19.1 | 22.9 | 107 | 124 | 17.3 | 116 |
| | RH | 85.2 | 86.1 | 20.4 | 24.0 | 111 | 131 | 20.1 | 118 |
| | SF | 83.6 | 82.1 | 27.0 | 32.3 | 120 | 153 | 33.1 | 128 |

**Table S3:** Summary Statistics for the Daily Maximum 8-Hour (MDA8) $O_3$ over the globe other regions, accompanying Table 2. The last two columns indicate the difference between the 99[th] percentile and the 90[th] percentile, expressed both in absolute values (ppb) and as a percent.

**Supplemental Description of the Super-Fast Chemical Mechanism**

The SF mechanism is in the CESM code archive as an unsupported chemical mechanism, which can be activated using the option '-chem super_fast_llnl'. 
[revised manuscript text omitted]

---

## Author Response (AR2)

Response to Reviewers II

Fiona O'Connor,

Thank you for the time taken to manage the review of our manuscript (https://www.geosci-model-dev-discuss.net/gmd-2018-16/).

We would like to extend our previous response to the additional simulations requested by Reviewer #2. The Super-Fast simulations requested (a set of simulations using ACCMIP pre-industrial emissions to compare with present-day) have already been conducted as part of the ACCMIP intercomparison itself, and have been included in multiple ACCMIP papers, including Young et al. (2013) (https://www.atmos-chem-phys.net/13/2063/2013/), among other model intercomparison publications. The conclusion from ACCMIP was that the performance and sensitivity of the Super-Fast mechanism is generally comparable to the other mechanisms in ACCMIP.  As such, we feel that the validity of the Super-Fast mechanism for the historical period has already been demonstrated and published.

In addition, we feel that the primary goal of our manuscript is not to validate the Super-Fast mechanism beyond what has been done with the ACCMIP publications, but rather to demonstrate how much is lost scientifically (and gained computationally) by utilizing chemical mechanisms of different complexities within a unified GCM framework. While additional historical simulations would be valuable, they are not critical for this particular manuscript and would be more appropriate for follow up studies.

To this point, we have updated the manuscript, primarily Section 2.2.3, to extend the introduction and description of the Super-Fast mechanism. We have moved the paragraph at the end of the section to the beginning, and added additional references and text, as well as a clear indication that there is an extended discussion of the Super-Fast mechanism in the Supplemental Material.

We have also added additional text and citations describing the performance of the Super-Fast mechanism within model intercomparisons to the Supplemental Material.

Please find included the updated manuscript and supplement in final form (pdf and docx) and as docx files with tracked changes.

We hope that this clarifies our position.

We look forward to your response.

Benjamin Brown-Steiner
Noelle Selin
Ronald Prinn
Louisa Emmons
Simone Tilmes
Jean-Francois Lamarque

Philip Cameron-Smith

[revised manuscript text omitted]
$_∞$))·0.6·exp((1//(1+log(ko_m/k$_∞$))$^2$)))+(k 0/(1+(k0/k$_∞$_m)))·0.6·exp(( (1/(1+(log(k0/k$_∞$_m))$^2$))) | simplified: no $CO_2$, $H → HO_2$, see note A |
| (11) | $CH_2O + OH$ | → | $CO + H_2O + HO_2$ | 5.50E-12·exp(125/T) | rates identical, simplified: $H → HO_2$ |
| (12) | $CH_3O_2 + HO_2$ | → | $CH_3OOH + O_2$ | 4.10E-13·exp(750/T) | identical |
| (13a) | $CH_3OOH + OH$ | → | $CH_3O_2 + H_2O$ | 2.70E-12·exp(200/T) | in combination, equivalent |
| (13b) | $CH_3OOH + OH$ | → | $CH_3O + H_2O + OH$ | 1.10E-12·exp(200/T) | |
| (14) | $CH_3O_2 + NO$ | → | $CH_2O + HO_2 + NO_2$ | 2.80E-12·exp(300/T) | identical |
| (15) | $CH_3O_2 + CH_3O_2$ | → | $2 \cdot CH_2O + 0.80 \cdot HO_2$ | 9.50E-14·exp(390/T) | different rates, simplified: 1 reaction instead of 2 |
| (16) | $H_2O + NO_2$ | → | $0.50 \cdot HNO_3$ | 4.00E-24 | no equivalent in MOZART, note B |
| (17a) | $DMS + OH$ | → | $SO_2$ | 1.100E-11·exp(-240/T) | |
| (17b) | $DMS + OH$ | → | $0.75 \cdot SO_2$ | 2.00E-10·exp(5820·[M])/ ((2.00E29/[O2]) + exp(6280·[M])) | different, see note C |
| (18) | $OH + SO_2 + M$ | → | $SO_4$ | ko=3.30E-31·(300/T)$^{4.30}$; ki=1.60E-12; f=0.60 | different, see note C |
| (19) | $H_2O_2 + SO_2$ | → | $SO_4$ | aqueous chemistry (see note D) | no equivalent, see note C |
| (20) | $O_3 + SO_2$ | → | $SO_4$ | aqueous chemistry (see note D) | no equivalent, see note C |
| (21a) | $ISOP + OH$ | → | $2 \cdot CH_3O_2$ | 2.70E-11·exp(390/T) | |
| (21b) | $ISOP + OH$ | → | $ISOP$ | 2.70E-11·exp(390/T) | different, see note E |
| (21c) | $ISOP + OH$ | → | $ISOP + 0.5 \cdot OH$ | 2.70E-11·exp(390/T) | |
| (22) | $ISOP + O_3$ | → | $.87 \cdot CH_2O + 1.86 \cdot CH_3O_2 + 0.06 \cdot HO_2 + 0.05 \cdot CO$ | 5.59E-15·exp(-1814/T) | different, see note E |

NOTES:

A: For rate: ko = 5.90E-33•(300/T)1.4; k$_∞$ = 1.10E-12•(T/300)1.3; ko_m = ko•[M]; k0 = 1.50E-13•(T/300)0.6; k$_∞$_m = (2.10E9•(T/300)6.1)/[M]

B: $HNO_3$ chemistry included only as reaction 8 and 16, with reaction 16 involving heterogeneous chemistry parameterization

C: DMS chemistry limited only to reaction with OH (reaction 17), $SO_4$ production simplified to reactions 18 with OH and 19 and 20 with aqueous chemistry (with a fixed pH in the cloud droplets)

D: Rate equations are included within the aerosol routines adapted from the MOZART-4 mechanism implimentation within CAM-chem

E: Isoprene chemistry paramaterized from UCI for ISOP + OH and from LLNL-IMAPCT for ISOP + $O_3$, see text for full details

**Table S2:** Full description of Super-Fast chemical mechanism as compared to the MOZART-4 mechanism of Emmons et al. (2010). Reaction rates are written out if they are of the Arrhenius form, or otherwise formulated. If the reaction rates are of the Troe form, they list the ko and ki parameters, as in Emmons et al. (2010). The simplifications made in the SF are noted by indicating what species is missing or modified when compared to Emmons et al. (2010). Chemical species are the same as in Emmons et al. (2010).

| | | [ppbv] Mean | [ppbv] Median | [ppbv] Standard Deviation | [%] Variability | [ppbv] 90th Percentile | [ppbv] 99th Percentile | [ppbv] 99th - 90th Percentile | [%] 99th - 90th Percentile |
|---|---|---|---|---|---|---|---|---|---|
| **W US** | **CASTNET** | 49.9 | 49.6 | 4.82 | 9.66 | 56.4 | 61.3 | 4.87 | 109 |
| | **MO** | 54.9 | 54.9 | 3.37 | 6.14 | 59.4 | 62.4 | 2.99 | 105 |
| | **RH** | 55.9 | 55.8 | 3.58 | 6.41 | 60.6 | 64.0 | 3.47 | 106 |
| | **SF** | 50.5 | 50.5 | 3.22 | 6.38 | 54.7 | 57.8 | 3.15 | 106 |
| **MW US** | **CASTNET** | 57.3 | 57.0 | 7.57 | 13.2 | 67.1 | 76.9 | 9.77 | 115 |
| | **MO** | 71.0 | 70.7 | 11.03 | 15.5 | 85.5 | 96.0 | 10.6 | 112 |
| | **RH** | 72.6 | 72.1 | 11.26 | 15.5 | 87.4 | 98.8 | 11.4 | 113 |
| | **SF** | 70.6 | 69.8 | 13.44 | 19.0 | 88.7 | 104 | 15.1 | 117 |
| **MW US single grid cell** | **CASTNET** | 62.4 | 60.6 | 14.5 | 23.2 | 81.7 | 104 | 22.7 | 128 |
| | **MO** | 84.6 | 83.8 | 16.2 | 19.2 | 106 | 126 | 19.4 | 118 |
| | **RH** | 86.2 | 85.3 | 15.8 | 18.4 | 107 | 126 | 19.5 | 118 |
| | **SF** | 85.4 | 82.5 | 22.4 | 26.2 | 116 | 149 | 32.8 | 128 |
| **SE US** | **CASTNET** | 51.5 | 51.3 | 8.18 | 15.9 | 62.3 | 71.9 | 9.64 | 115 |
| | **MO** | 60.2 | 59.5 | 9.22 | 15.3 | 72.5 | 83.7 | 11.2 | 115 |
| | **RH** | 61.3 | 60.5 | 10.1 | 16.6 | 74.9 | 87.9 | 12.9 | 117 |
| | **SF** | 58.6 | 57.8 | 11.4 | 19.4 | 73.9 | 87.1 | 13.2 | 118 |
| **SE US single grid cell** | **CASTNET** | 53.8 | 53.1 | 12.8 | 23.8 | 68.7 | 93.5 | 24.8 | 136 |
| | **MO** | 83.4 | 84.7 | 19.1 | 22.9 | 107 | 124 | 17.3 | 116 |
| | **RH** | 85.2 | 86.1 | 20.4 | 24.0 | 111 | 131 | 20.1 | 118 |
| | **SF** | 83.6 | 82.1 | 27.0 | 32.3 | 120 | 153 | 33.1 | 128 |

**Table S3:** Summary Statistics for the Daily Maximum 8-Hour (MDA8) $O_3$ over the globe other regions, accompanying Table 2. The last two columns indicate the difference between the 99[th] percentile and the 90[th] percentile, expressed both in absolute values (ppb) and as a percent.

**Supplemental Description of the Super-Fast Chemical Mechanism**

The SF mechanism is in the CESM code archive as an unsupported chemical mechanism, which can be activated using the option '-chem super_fast_llnl'. The SF mechanism has been included in several model inter-comparison projects, including the ACCMIP (e.g. Lamarque et al., 2013a), a comparison of stratospheric dynamics and ozone production (Hsu et al., 2013), a comparison of isoprene mechanisms and ozone changes (Squire et al., 2015), and a multi-model assessment of surface ozone and observations (Schnell et al., 2015).  Here we briefly review the findings of these four model inter-comparison projects.

The SF only simulates sulfate ($SO_4$) and not the other aerosols, so the SF mechanism was not included in many of the ACCMIP aerosol comparisons (Lamarque et al., 2013a). While the inclusion of non-sulfate aerosols within the CESM can be easily accomplished, there are two aerosol modules (either bulk or modal) to which aerosols could be added, which was beyond the scope of this project, so aerosol model capabilities are not examined in the present study.

We now summarize the ACCMIP results as they pertain to the SF mechanism. Within the ACCMIP, the SF mechanism has lower rates of ozone production and loss compared to the ACCENT models (biases of -24% and -22% respectively), as well as low ozone deposition (bias of -38%) (Young et al., 2013). In this comparison, natural emissions were not prescribed and different treatments of meteorology were used, which may account for some of the noted differences. This results in a high bias for the ozone lifetime (+3 days, or +14%), as well as a low ozone burden bias (-34 Tg, or -10%) (Young et al., 2013). In addition, the models that showed similarly low ozone production and loss rates have lower emissions of VOCs. The SF mechanism falls within the ACCMIP range for human health results due to ozone exposure (Silva et al., 2013). The SF mechanism simulated the historical and future changes in the tropospheric ozone column and radiative forcing within the range of the other ACCMIP models (Stevenson et al., 2013), with SF being near the end of the range that simulated larger changes.  The SF mechanism also has a high bias for global-mean OH (+16% compared to the ACCMIP mean) and a low bias for the calculation of the methane lifetime due to OH oxidation (-14%) (Voulgarakis et al., 2013).

The SF mechanism was tested against MOZART by Hsu et al. (2013) who concluded that the selection of a chemical mechanism was only a secondary influence on the stratospheric chemistry since they used a linearized scheme. However, the SF mechanism did produce a less stratified tropopause and a warmer troposphere due largely to the impact of ozone forcings on the simulated dynamics and thermodynamics. Unfortunately, the Hsu et al. (2013) analysis had a bug with their SF simulations, which resulted in the aerosols not being communicated to the cloud nucleation routines, but this didn't affect their conclusions on the sensitivity of the stratosphere to uncertainty in the $O_2$ photolysis cross-section.

Squire et al. (2015) compared the SF isoprene scheme with three other schemes of much greater complexity. They concluded that the "1-species, 2-reaction" isoprene scheme from the SF mechanism, as simple as it is, is preferable to neglecting biogenic chemistry entirely, although the SF mechanism shows the highest biases in regions where isoprene chemistry is important for simulating accurate ozone concentrations. They also explored some of the other biases within the SF mechanism scheme, which include: (1) under high-isoprene conditions, the SF mechanism overestimates $O_3$; (2) under low-isoprene and low-$NO_x$ conditions, the SF mechanism overestimates $O_3$; (3) due to the simplicity of SF mechanism, $HO_x$ is sequestered into the organic hydroperoxides, and methyl hydroperoxide ($CH_3OOH$) has low reactivity, which results in high levels of the peroxy radicals, an enhanced rate of $CH_3O_2 + NO$, and therefore a high bias (up to +80%) for ozone; and (4) the $NO_x$ lifetime is too short, except in high-$NO_x$ emission regions. They conclude that the addition of a PAN formation scheme would significantly improve the $O_3$ distribution. Finally, they find that many of the errors described above largely cancel each other out, which results in the globally averaged $O_3$ bias for SF mechanism to be small (-2.6% compared to the Master Chemical Mechanism).

The SF mechanism has a known anomalous annual cycle (see Schnell et al., 2015), in which peak ozone occurs in March/April rather than May. In the main article we show that this anomaly exists at global scales, but not within all regions. In addition, the size and extent of ozone pollution episodes is anomalously high, and these large events occur mainly in the springtime (Schnell et al., 2015). Interestingly, the SF mechanism outperforms many of the more sophisticated mechanisms in simulating the observed summertime diurnal cycle for ozone (Schnell et al., 2015).

Other studies that have used SF, or intercompare its results with other atmospheric chemistry models, include: ozone impacts in the Coupled Model Intercomparison Project Phase 5 (CMIP5) (Eyring, et al., 2013); air pollution and mortality (Silva et al., 2016, 2017); the role of dimethyl sulphide (DMS) in climate  (Xu et al., 2016; Wang et al., 2018a, 2018b), and other ACCMIP related papers (Fiore et al., 2012; Bowman, et al., 2013; Naik, et al., 2013, Lamarque, et al., 2013b; Kirschke, et al., 2013).  Overall, SF produced simulations that were generally acceptable.

**References for Supplemental Material:**

Bowman, K. W., Shindell, D. T., Worden, H. M., Lamarque, J. F., Young, P. J., Stevenson, D. S., Qu, Z., de la Torre, M., Bergmann, D., Cameron-Smith, P. J., Collins, W. J., Doherty, R., Dalsøren, S. B., Faluvegi, G., Folberth, G., Horowitz, L. W., Josse, B. M., Lee, Y. H., MacKenzie, I. A., Myhre, G., Nagashima, T., Naik, V., Plummer, D. A., Rumbold, S. T., Skeie, R. B., Strode, S. A., Sudo, K., Szopa, S., Voulgarakis, A., Zeng, G., Kulawik, S. S., Aghedo, A. M., and Worden, J. R.: Evaluation of ACCMIP outgoing longwave radiation from tropospheric ozone using TES satellite observations, Atmos. Chem. Phys., 13, 4057-4072, https://doi.org/10.5194/acp-13-4057-2013, 2013.

Emmons, L. K., Walters, S., Hess, P. G., Lamarque, J.-F., Pfister, G. G., Fillmore, D., Granier, C., Guenther, A., Kinnison, D., Laepple, T., Orlando, J., Tie, X., Tyndall, G., Wiedinmyer, C., Baughcum, S. L., and Kloster, S.: Description and evaluation of the Model for Ozone and Related chemical Tracers, version 4 (MOZART-4), Geosci. Model Dev., 3, 43-67, https://doi.org/10.5194/gmd-3-43-2010, 2010.

Eyring, V., Arblaster, J. M., Cionni, I., Sedláček, J., Perlwitz, J., Young, P. J., Bekki, S., Bergmann, D., Cameron-Smith, P., Collins, W. J., Faluvegi, G., Gottschaldt, K.-D., Horowitz, L. W., Kinnison, D. E., Lamarque, J.-F., March, D. R., Saint-Martin, D., Shindell, D. T., Sudo, K., Szopa, S., Watanabe, S.: Long-term ozone changes and associated climate impacts in CMIP5 simulations, J. Geophys. Res.-Atmos., 118, 5029-5060, doi:10.1002/jgrd.50316, 2013.

Fiore, A. M., Naik, V., Spracklen, D. V., Steiner, A., Unger, N., Prather, M., Bergmann, D., Cameron-Smith, P. J., Cionni, I., Collins, W. J., Dalsoren, S., Eyring, V., Folberth, G. A., Ginoux, P., Horowitz, L. W., Josse, B., Lamarque, J.-F., MacKenzie, I. A., Nagashima, T., O'Connor, F. M., Righi, M., Rumbold, S. T., Shindell, D. T., Skeie, R. B., Sudo, K., Szopa, S., Takemura, T., and Zeng, G.: Global air quality and climate, Chem. Soc. Rev., 41, 6663–6683, doi:10.1039/c2cs35095e, 2012.

Hsu, J., Prather, M. J., Bergmann, D., and Cameron-Smith, P.: Sensitivity of stratospheric dynamics to uncertainty in $O_3$ production, J. Geophys. Res. Atmos., 118, 8984-8999, https://doi.org/10.1002/jgrd.50689, 2013.

Kirschke, S., Bousquet, P., Ciais, P., Saunois, M., Canadell, J. G., Dlugokencky, E. J., Bergamaschi, P., Bergmann, D., Blake, D.R., Bruhwiler, L., Cameron-Smith, P., Castaldi, S., Chevallier,F., Feng, L., Fraser, A., Heimann, M., Hodson, E. L., Houwel-ing, S., Josse, B., Fraser, P. J., Krummel, P. B., Lamarque, J.-F., Langenfelds, R. L., Le Quéré, C., Naik, V., O'Doherty, S.,Palmer, P. I., Pison, I., Plummer, D., Poulter, B., Prinn, R. G.,Rigby, M., Ringeval, B., Santini, M., Schmidt, M., Shindell, D.T., Simpson, I. J., Spahni, R., Steele, L. P., Strode, S. A., Sudo,K., Szopa, S., van der Werf, G.R., Voulgarakis, A., van Weele,M., Weiss, R. F., Williams, J. E., and Zeng, G.: Three decadesof global methane sources and sinks, Nat. Geosci., 6, 813–823, doi:10.1038/ngeo1955, 2013.

Lamarque, J.-F., Shindell, D. T., Josse, B., Young, P. J., Cionni, I, Eyring, V., Bergmann, D., Cameron-Smith, P., Collins, W. J., Doherty, R., Dalsoren, S., Faluvegi, G., Folberth, G., Ghan, S. J., Horowitz, L. W., Lee, Y. H., MacKenzie, I. A., Nagashima, T., Naik, V., Plummer, D., Righi, M., Rumbold, S. T., Schulz, M., Skeie, R. B., Stevenson, D. S., Strode, S., Sudo, K., Szopa, S., Voulgarakis, A., and Zeng, G.: The atmospheric chemistry and climate model intercomparison Project (ACCMIP): Overview and description of models, simulations and climate diagnostics, Geosci. Mod. Dev., 6, 179–206, https://doi.org/10.5194/gmd-6-179-2013, 2013a.

Lamarque, J.-F., Dentener, F., McConnell, J., Ro, C.-U., Shaw, M., Vet, R., Bergmann, D., Cameron-Smith, P., Dalsoren, S., Doherty, R., Faluvegi, G., Ghan, S. J., Josse, B., Lee, Y. H., MacKenzie, I. A., Plummer, D., Shindell, D. T., Skeie, R. B., Stevenson, D. S., Strode, S., Zeng, G., Curran, M., Dahl-Jensen, D., Das, S., Fritzsche, D., and Nolan, M.: Multi-model mean nitrogen and sulfur deposition from the Atmospheric Chemistry and Climate Model Intercomparison Project (ACCMIP): evaluation of historical and projected future changes, *Atmos. Chem. Phys.*, **13**, 7997-8018, doi:10.5194/acp-13-7997-2013, 2013b.

Naik, V., Voulgarakis, A., Fiore, A. M., Horowitz, L. W., Lamarque, J.-F., Lin, M., Prather, M. J., Young, P. J., Bergmann, D., Cameron-Smith, P. J., Cionni, I., Collins, W. J., Dalsøren, S. B., Doherty, R., Eyring, V., Faluvegi, G., Folberth, G. A., Josse, B., Lee, Y. H., MacKenzie, I. A., Nagashima, T., van Noije, T. P. C., Plummer, D. A., Righi, M., Rumbold, S. T., Skeie, R., Shindell, D. T., Stevenson, D. S., Strode, S., Sudo, K., Szopa, S., and Zeng, G.: Preindustrial to

present-day changes in tropospheric hydroxyl radical and methane lifetime from the Atmospheric Chemistry and Climate Model Intercomparison Project (ACCMIP), Atmos. Chem. Phys., 13, 5277-5298, https://doi.org/10.5194/acp-13-5277-2013, 2013.

Schnell, J. L., Prather, M. J., Josse, B., Naik, V., Horowitz, L. W., Cameron-Smith, P., Bergmann, D., Zeng, G., Plummer, D. A., Sudo, K., Nagashima, T., Shindell, D. T., Faluvegi, G., and Strode, S. A.: Use of North American and European air quality networks to evaluate global chemistry-climate modeling of surface ozone, Atmos. Chem. Phys., 15, 10581–10596, https://doi.org/10.5194/acp-15-10581-2015, 2015.

Silva, R. A., West, J. J., Zhang, Y., Anenberg, S. C., Lamarque, J.-F., Shindell, D. T., Collins, W. J., Dalsoren, S., Faluvegi, G., Folberth, G., Horowitz, L. W., Nagashima, T., Naik, V., Rumbold, S., Skeie, R., Sudo, K., Takemura, T., Bergmann, D., Cameron-Smith, P., Cionni, I., Doherty, R. M., Eyring, V., Josse, B., MacKenzie, I. A., Plummer, D., Righi, M., Stevenson, D. S., Strode, S., Szopa, S., and Zeng, G.: Global premature mortality due to anthropogenic outdoor air pollution and the contribution of past climate change, Environ. Res. Lett., 8, 034005, https://doi.org/10.1088/1748-9326/8/3/034005, 2013.

Silva, R. A., West, J. J., Lamarque, J.-F., Shindell, D. T., Collins, W. J., Dalsoren, S., Faluvegi, G., Folberth, G., Horowitz, L. W., Nagashima, T., Naik, V., Rumbold, S. T., Sudo, K., Takemura, T., Bergmann, D., Cameron-Smith, P., Cionni, I., Doherty, R. M., Eyring, V., Josse, B., MacKenzie, I. A., Plummer, D., Righi, M., Stevenson, D. S., Strode, S., Szopa, S., and Zengast, G.: The effect of future ambient air pollution on human premature mortality to 2100 using output from the ACCMIP model ensemble, Atmos. Chem. Phys., 16, 9847-9862, doi:10.5194/acp-16-9847-2016, 2016.

Silva, R. A., West, J. J., Lamarque, J.-F., Shindell, D. T., Collins, W. J., Faluvegi, G., Folberth, G. A., Horowitz, L. W., Nagashima, T., Naik, V., Rumbold, S. T., Sudo, K., Takemura, T., Bergmann, D., Cameron-Smith, P., Doherty, R. M., Josse, B., MacKenszie, I. A., Stevenson, D. S., and Zeng, G.: Future global mortality from changes in air pollution attributable to climate change, Nature Climate Change, 7, 647-651, doi:10.1038/nclimate3354, 2017.

Squire, O. J., Archibald, A. T., Griffiths, P. T., Jenkin, M. E., Smith, D., and Pyle, J. A.: Influence of isoprene chemical mechanism on modelled changes in tropospheric ozone due to climate and land use over the 21st century, Atmos. Chem. Phys., 15, 5123–5143, https://doi.org/10.5194/acp-15-5123-2015, 2015.

Stevenson, D. S., Young, P. J., Naik, V., Lamarque, J.-F., Shindell, D. T., Voulgarakis, A., Skeie, R. B., Dalsøren, S. B., Myhre, G., Berntsen, T. K., Folberth, G. A., Rumbold, S. T., Collins, W. J., MacKenzie, I. A., Doherty, R. M., Zeng, G., van Noije, T. P. C., Strunk, A., Bergmann, D., Cameron-Smith, P., Plummer, D. A., Strode, S. A., Horowitz, L., Lee, Y. H., Szopa, S., Sudo, K., Nagashima, T., Josse, B., Cionni, I., Righi, M., Eyring, V., Conley, A., Bowman, K. W., and Wild, O.: Tropospheric ozone changes, radiative forcing and attribution to emissions in the Atmospheric Chemistry and Climate Model Intercomparison Project (ACCMIP), Atmos. Chem. Phys., 13, 3063–3085, https://doi.org/10.5194/acp-13-3063-2013, 2013.

Voulgarakis, A., Naik, V., Lamarque, J.-F., Shindell, D. T., Young, P. J., Prather, M. J., Wild, O., Field, R. D., Bergmann, D., Cameron-Smith, P., Cionni, I., Collins, W. J., Dalsøren, S. B., Doherty, R. M., Eyring, V., Faluvegi, G., Folberth, G. A., Horowitz, L. W., Josse, B., MacKenzie, I. A., Nagashima, T., Plummer, D. A., Righi, M., Rumbold, S. T., Stevenson, D. S., Strode, S. A., Sudo, K., Szopa, S., and Zeng, G.: Analysis of present day and future OH and methane lifetime in the ACCMIP simulations, Atmos. Chem. Phys., 13, 2563–2587, 2013.

Wang, S., Maltrud, M., Burrows, S., Elliott, S., and Cameron-Smith, P.: Impacts of shifts in phytoplankton community on clouds and climate via the sulfur cycle, Global Biogeochemical Cycles, 32, 6 1005-1026, doi:10.1029/2017GB005862, 2018a.

Wang, S., Maltrud, M., Elliott, S., Cameron-Smith, P., and Jonko, A.: Influence of dimethyl sulfide on the carbon cycle and biological production, Biogeochemistry, 138, 1, 49-68, doi:10.1007/s10533-018-0430-5, 2018b.

Xu, L., Cameron-Smith, P., Russell, L. M., Ghan, S. J., Liu, Y., Elliott, S., Yang, y., Lou, S., Lamjiri, M. A., and Manizza, M.: DMS role in ENSO cycle in the tropics, J. Geophys. Res.-Atmos., 121, 537–558, https://doi.org/10.1002/2016JD025333, 2016.

Young, P. J., Archibald, A. T., Bowman, K. W., Lamarque, J.-F., Naik, V., Stevenson, D. S., Tilmes, S., Voulgarakis, A., Wild, O., Bergmann, D., Cameron-Smith, P., Cionni, I., Collins, W. J., Dalsøren, S. B., Doherty, R. M., Eyring, V., Faluvegi, G., Horowitz, L. W., Josse, B., Lee, Y. H., MacKenzie, I. A., Nagashima, T., Plummer, D. A., Righi, M., Rumbold, S. T., Skeie, R. B., Shindell, D. T., Strode, S. A., Sudo, K., Szopa, S., and Zeng, G.: Pre-industrial to end 21st century projections of tropospheric ozone from the Atmospheric Chemistry and Climate Model Intercomparison Project (ACCMIP), Atmos. Chem. Phys., 13, 2063–2090, https://doi.org/10.5194/acp-13-2063-2013, 2013.

---

## Author Response (AR3)

Thank you for your review and valuable comments. Please see our response below (reviewer comments in italics, our response will follow, with text modifications in bold).

*Page 13, line 28: Whilst I agree that there should be a benefit in running more simulations, if one is to look at parametric uncertainty in a highly parametrised scheme, what will we really learn about the atmosphere? I agree that this would be valuable for "making the model better" but are there not risks associated with this? Risks that we would end up over-tuning models, which would inevitably result in false confidence in their predictions? I would appreciate some comment on exactly how you see the benefits to our understanding of the atmosphere using ensembles of simulations with chemistry schemes that lack the sensitivity of the real atmosphere (i.e. are simplified -- you may like to rebut my assumption that simplified schemes have the incorrect sensitivity).*

The ability for simplified/parameterized models to inform and constrain more complex, less parameterized models is well demonstrated in the Earth system Models of Intermediate Complexity (EMIC) field, and part of the motivation for this work was to see how simplified global 3D chemistry models could stand in between 2D EMICs and full 3D complex climate-chemistry models. While there are risks of over tuning models, the potential benefit in providing a much broader context of the larger parameter space we feel is quite large. As all mechanisms are simplifications of the real atmosphere, the issue of complexity is largely a matter of degree. We have mentioned in this manuscript the need for both simple and complex models to be used in parallel, and we feel that maintaining a connection between the characterizations in the simplified models and the complex models is of critical importance. To this point, we've added the following (red text) in the manuscript and trimmed the beginning of the following sentence to incorporate this addition.

**Page 13, Line 28: We feel that the capability to run three SF simulations for the price of one MO simulation under different sets of initial conditions, for example, can extend the quantification of parametric uncertainties which is largely unavailable to the most complex and most computationally demanding mechanisms. Of course, the SF mechanism may not be appropriate for every sensitivity study, but neither is the MO mechanism. The choice of mechanism really depends, then, on the science question. If the research objective is to predict complex chemistry-climate interactions and computational resources are available, then a more complex mechanism will have the most value. However, if the research objective is to better understand various parameterizations, then a more computationally efficient mechanism will have higher value, even if it might not be capable of accurately simulating all variables in detail (Hoffman et al., 1996). This is particularly the case when a baseline can be established between the simplified mechanism and the complex mechanism, as we have done here. We feel that this parallel approach, in which a set of mechanisms with varying levels of complexity are run concurrently with a consistent set of parameters, allows us to enhance our exploration of uncertainties and thus our ability to understand the atmospheric chemistry of the Earth system.**
**For instance, there are many research frameworks where the "three-for-one" advantage of the SF mechanism could be utilized with the MO mechanism in which one simulation…**

Hoffman, R., Minkin, V. I., and Carpenter, B. K.: Ockham's Razor and Chemistry, Bull. Soc. Chim. France, 133, 117-130, 1996.

*Page 15, line 31: "efficiency" should, I think, be "efficiently".*

Corrected.